

# A geometry-dependent surface Lambertian-equivalent reflectivity product for UV/Vis retrievals: Part II. Evaluation over open ocean

Zachary Fasnacht[1], Alexander Vasilkov[1], David Haffner[1], Wenhan Qin[1], Joanna Joiner[2], Nickolay Krotkov[2], Andrew M. Sayer[3], and Robert Spurr[4]

[1]Science Systems and Applications Inc., Lanham, MD, USA
[2]NASA Goddard Space Flight Center, Greenbelt, MD, USA
[3]GESTAR, Universities Space Research Association, Columbia, MD, USA
[4]RT Solutions Inc., Cambridge, MA, USA

*Correspondence to:* Z. Fasnacht (zachary.fasnacht@ssaihq.com)

**Abstract.** Satellite-based cloud, aerosol, and trace gas retrievals from ultraviolet (UV) and visible (Vis) wavelengths depend on the accurate representation of surface reflectivity. Current UV and Vis retrieval algorithms typically use surface reflectivity climatologies that do not account for variation in satellite viewing geometry or surface roughness. The concept of geometry-dependent surface Lambertian-equivalent reflectivity (GLER) is implemented for water surfaces to account for surface anisotropy using a Case 1 water optical model and the Cox-Munk slope distribution for ocean surface roughness. GLER is compared with Lambertian-Equivalent reflectivity (LER) derived from the Ozone Monitoring Instrument (OMI) for clear scenes at 354, 388, 440, and 466 nm. We show that GLER compares well with the measured LER data over the open ocean and captures the directionality effects not accounted for in climatological LER databases. Small biases are seen when GLER and the OMI-derived LER are compared. GLER is biased low by up to 0.01-0.02 at Vis wavelengths, and biased high by around 0.01 in the UV, particularly at 354 nm. Our evaluation shows that GLER is an improvement upon climatological LER databases as it compares well with OMI measurements, and captures the directionality effects of surface reflectance.

## 1 Introduction

Satellite retrievals of clouds, aerosols, and trace gases rely on the accurate representation of surface reflectivity. Many modern satellite ultraviolet (UV) and visible (Vis) trace gas algorithms use the mixed Lambert equivalent reflectivity (LER) model which assumes the measured top-of-atmosphere (TOA) radiance is a combination of the clear and cloudy sky radiances weighted by an effective





cloud fraction (ECF) (Koelemeijer et al., 2001; Seftor et al., 1994; Stammes et al., 2008). While many databases of LER currently exist, they are typically climatological LER data sets based on the minimum or the mode of the LER distribution. Therefore, they do not account for the directionality effects due to the variation of satellite viewing and solar illumination geometries. Further, they

do not capture day to day change in the roughness of the water surface due changing wind speed which can impact surface reflectance. Additionally, climatological LER databases may be affected by residual aerosol and cloud contamination. These databases include Kleipool et al. (2008) using data from the Ozone Monitoring Instrument (OMI) for wavelengths 328-499 nm, Koelemeijer et al. (2003) from the Global Ozone Monitoring Experiment (GOME) for wavelengths 335-772 nm, and

Tilstra et al. (2017) from GOME-2 for wavelengths between 335-772 nm.

The bidirectional reflectance distribution function (BRDF) describes the reflectivity of a surface for all illumination and viewing geometries. Given a specific solar and viewing from a satellite and the BRDF of the surface, a quantity known as the bidirectional reflectance factor (BRF) can be derived for that surface. BRF is defined as the ratio of the radiant flux reflected by a surface to

the radiant flux reflected into the identical beam geometry by an ideal diffuse Lambertian surface, irradiated under the same conditions as the sample surface (Schaepman-Strub et al., 2006). A peak in the BRDF arises due to sun glint phenomena when the the sun and satellite viewing angles are similar and oriented in a forward scattering geometry. Sun glint is strongest for smooth water surfaces that permit nearly perfect Fresnel reflection of direct light from the ocean surface (Kay et al., 2009; Cox

and Munk, 1954; Thomas et al., 2012). Another feature of the BRDF over water occurs near the outer edges of satellite swath viewing geometry, when TOA radiances increase due to increased effects of Rayleigh scattering. The increase in diffuse sky reflection from the ocean surface becomes more significant at these longer path lengths, and the relative contributions from the water leaving radiance to the intensity reaching the satellite is reduced (Vasilkov et al., 2017).

Vasilkov et al. (2017) introduced a concept known as geometry-dependent LER (GLER), where for a specific viewing geometry, TOA radiances are simulated over a non-Lambertian surface using the BRDF. GLER can be easily implemented into trace gas algorithms by simply replacing the currently used LER climatologies. For Vis wavelengths, the change in surface reflectivity associated with implementation of GLER was found to directly decrease the OMI $NO_2$ air mass factor (AMF)

over land and oceans by as much as 15%, and change the OMI $NO_2$ AMF indirectly by an additional $-22\%$ to 13% through changes to retrieved cloud properties (Vasilkov et al., 2017). In the UV, Ahn et al. (2014) noted that unrealistic surface albedo is one of the main causes for uncertainty in aerosol retrievals, while Torres et al. (1998) reported that surface albedo errors can lead to as much as a 5% error in retrieved aerosol optical depth (AOD) for weakly absorbing aerosols. Lee et al. (2009)

showed that for clear scenes, errors in surface reflectivity in the UV can lead to as much as an 8% error in the calculation of $SO_2$ AMF over oceans.

Qin et al. (2019) evaluated the 466 nm GLER product over land generated using BRDF data



from the Moderate Resolution Imaging Spectroradiometer (MODIS). They compared GLER with OMI-derived LER globally and found GLER was biased low by 0.01 to 0.02 relative to OMI. The difference was attributed to several factors, including small calibration differences between MODIS and OMI and possible residual cloud and/or aerosol contamination in the OMI data that was not
completely filtered out.

Here we evaluate the GLER product generated for the OMI instrument for ocean scenes. OMI is a hyperspectral UV-Vis (270-500 nm) imager onboard the NASA Aura satellite, which was launched in July 2004. The high spectral resolution (0.42-0.63 nm) of the UV (270-370 nm) and Vis (350-500 nm) channels enables retrievals of many important atmospheric constituents including $O_3$, $NO_2$,
$SO_2$, and aerosols. OMI has a spatial resolution of 13 km x 24 km at nadir with a field of view (FOV) of 0.8° in the flight direction and 115° across the swath. Prior to 2008, OMI provided global coverage daily with a repeat cycle of 16 days. The OMI row anomaly affects data coverage starting in mid-2007 (Schenkeveld et al., 2017), but a substantial amount of high quality global data remain available thereafter.

In this work we focus on evaluation of GLER at UV and Vis wavelengths over oceans. In the UV, 354 nm and 388 nm are chosen for evaluation as both have importance in the OMI retrieval of aerosol properties (Torres et al., 2007) and additionally OMI Raman cloud retrievals are performed at 354 nm (Vasilkov et al., 2008). For Vis wavelengths, 440 nm and 466 nm are chosen as they are important for $O_2$-$O_2$ cloud retrievals (Vasilkov et al., 2017; Veefkind et al., 2016) as well as $NO_2$
retrievals (Krotkov et al., 2017; Lamsal et al., 2014).

Whereas the land product described in Qin et al. (2019) used a model of BRDF with input from MODIS, GLER for ocean scenes is produced solely by modeling of water-leaving radiance and surface reflection. These surface-leaving radiance contributions are geometry-dependent, and the anisotropic nature of light backscattered by the ocean has been studied in many papers [see e.g.,
Gordon (1989); Morel and Gentili (1991, 1993, 1996); Park and Ruddick (2005); Lee et al. (2013)]. For GLER, water-leaving radiance is simulated using a Case 1 water model (Morel, 1988; Morel and Maritorena, 2001) that depends only on chlorophyll concentration. This model is described in detail in Sect. 2.2. Reflection from the ocean surface is modeled using the Cox-Munk slope distribution (Cox and Munk, 1954) and is further described in Sect. 2.3.

The algorithms and approaches described in this paper are relevant to NASA's future Plankton, Aerosol, Cloud, ocean Ecosystem mission (PACE) mission. PACE is currently is planned to launch in 2022-2023. Global PACE observations will provide data to monitor oceanic and atmospheric variables important for Earth ecosystem, carbon cycle, and climate studies. The PACE Ocean Color Instrument (OCI) is designed as a wide-swath imaging spectrometer with a 1 km ground nadir reso-
lution, a 5 nm spectral resolution between 345 and 890 nm and several short wave infrared bands. As compared with the Sea-Viewing Wide Field-of-View Sensor (SeaWiFS), the MODerate resolution Imaging Spectrometer (MODIS), and the Visible Infrared Imaging Radiometer Suite (VIIRS), the


OCI will additionally measure TOA radiances in the UV to help identify phytoplankton composition and harmful algal blooms. PACE's spectral coverage from UV-A to green wavelength region and in the red-NIR will enable unparalleled evaluation of ocean ecosystem properties in optically complex waters and in regions of increasing eutrophication (Cetinic et al., 2018).

## 5 2   Data and Methods

### 2.1   VLIDORT Radiative Transfer Model

For radiative transfer calculations we use the Vector Linearized Discrete Ordinate Radiative Transfer (VLIDORT) model. VLIDORT is a multiple scattering radiative transfer model that can simulate Stokes vectors at any level in the atmosphere and for any scattering geometry with a Lambertian

or non-Lambertian underlying surface (Spurr, 2006; Spurr et al., 2019). VLIDORT can simulate attenuation of solar and line-of-sight paths in a spherical atmosphere. In this study, we correct for the effects of atmospheric sphericity for both incoming solar and outgoing viewing directions based on a regular pseudo-spherical geometry calculation. This is important for large solar and viewing zenith angles. We also include polarization using the vector mode, because to neglect it can lead to

considerable errors for modeling backscattered radiances in the UV/Vis wavelength range.

### 2.2   Water-Leaving Radiance Implementation

VLIDORT has a supplement ("VSLEAVE") for the generation of surface-leaving radiances for use as inputs to the main radiative transfer calculation in the atmosphere. This supplement can be used for either simulations of solar-induced fluorescence or water-leaving radiances from Case 1 waters.

Our Case 1 water model accounts for the bi-directional effects following Morel and Gentili (1996). In this paper we do not account for vibrational-Raman scattering in ocean water (Morel et al., 2002; Vasilkov et al., 2002). The common Case 1 water model developed for the Vis (Morel, 1988) was extended to the UV using a parameterization of the particulate matter absorption coefficient from Vasilkov et al. (2002, 2005). The model requires as input several quantities that affect absorption

and scattering properties of sea-water and its constituents. Extinction coefficients for water absorption are taken from Lee et al. (2015), chlorophyll absorption coefficients from Vasilkov et al. (2005) below 400 nm and Lee et al. (2005) at longer wavelengths, and CDOM absorption from Morel and Maritorena (2001). The water scattering coefficients we use are from Morel et al. (2007) and for chlorophyll scattering, Morel and Maritorena (2001). A detailed description of these parameteriza-

tions is provided in Appendix A2.

The computation of emerging water-leaving radiance $L_w$ depends not only on the optical properties of marine constituents and radiative processes in the near-surface ocean, but also on the total atmospheric direct and diffuse downwelling transmittance $T_{atm}$ of atmospheric light through the





air-water interface. This complicates separation of the water-leaving calculation and the calculation of atmospheric radiance propagation in the atmosphere. Additionally, $T_{atm}$ will in general depend on the surface leaving contribution and hence on marine constituents. VLIDORT and its supplement VSLEAVE are therefore coupled. This coupling can be treated formally with a coupled ocean-

atmosphere radiative transfer model such as that described in Spurr et al. (2007). Here, however, we have developed a simple coupling scheme for VLIDORT that ensures the value of $L_w$ used as a input at the ocean surface will correspond to the correct value of the downwelling flux reaching the surface interface. The first applications of this new water-leaving model were presented in Vasilkov et al. (2017) and Sayer et al. (2017). The coupled model approach is described further in Appendix

B.

### 2.3   Cox-Munk BRDF Implementation

A supplement ("VBRDF") is implemented in VLIDORT to account for the reflection of the water surface using the Cox-Munk slope distribution (Cox and Munk, 1954). We use the full form of the Cox-Munk distribution in which the facet slope variance depends on both wind speed and wind

direction. Polarization at the ocean surface is accounted for using a full Fresnel reflection matrix as suggested by Mishchenko and Travis (1997). Additionally we account for contributions from oceanic foam that can be significant for high wind speeds using work by Frouin et al. (1996).

### 2.4   Ancillary Data for Water Model

As mentioned in the introduction, modeling of the water leaving radiance requires information on

the chlorophyll concentration and the modeling of Cox-Munk surface roughness depends on wind speed and wind direction. These inputs are not available directly from the OMI satellite and so other sources of ancillary inputs are required.

   The wind speed measurements for GLER come from a pair of satellite microwave imagers. Wind speed data are from the Advanced Microwave Scanning Radiometer - Earth Observing System

(AMSR-E) instrument onboard the NASA Aqua satellite with a spatial resolution of $0.25°$ (Wentz and Meissner, 2004). The AMSRE-E instrument, however, ceased operations in October 2011 due to an issue with the spinning mechanism (Wentz and Meissner, 2007). After October 2011, data are taken from the Special Microwave Imager/Sounder (SSMIS) with a spatial resolution of $0.25°$. SSMIS is onboard the Air Force Defense Meteorological Satellite Program (DMSP) Satellite F16

(Wentz, et al., 2012). While the SSMIS instrument has been operating since October 2003, the AMSR-E instrument was chosen for the first half of the OMI mission due to the small difference in equator crossing times of 7-15 minutes between the Aqua and Aura satellites, wheres the F16 satellite crossing times range from 6 hours behind Aura in 2005 to 2 hours behind Aura currently.





In future work, we plan to replace the SSMIS F16 wind speed with the AMSR-2 wind speed data. AMSR-2 is on board the Global Change Observation Mission Water Satellite 1 (GCOM-W1) which has an equator crossing time more similar to OMI. Gaps in the wind speed data due to extreme glint or rain are filled by the Global Modeling Assimilation Office (GMAO) Goddard Earth Observing

System Model Forward Processing for Instrument Teams (GEOS-5 FP-IT) near real-time assimilation with a spatial resolution of $0.625°$ longitude by $0.5°$ latitude and a temporal resolution of 3 hours (Lucchesi, 2013). Wind direction data are also from the GEOS-5 FP-IT model.

Monthly chlorophyll data from the MODIS instrument which is on board NASA Aqua satellite were used in modeling of the water leaving radiance (Hu et al., 2012). These chlorophyll data have a

spatial resolution of 4 km. The MODIS daily chlorophyll data are not used due to large gaps caused by clouds and aerosols. Some gaps still exist in the monthly data and are filled by other data sets from the MODIS team. These include a monthly climatological and yearly chlorophyll data set which are also at 4 km resolution. The benefit of using the monthly chlorophyll data instead of the climatological data comes from the ability to capture inter-annual trends due to phenomena such as

the El Niño-Southern Oscillation (ENSO).

## 2.5   Calculation of LER and GLER

Using the equation from Dave (1978), LER can be calculated from TOA Radiance ($I_{comp}$) by inverting the following:

$$I_{\text{comp}}(\lambda, \theta, \theta_0, \phi, P_s, \text{BRF}_s) \approx I_0(\lambda, \theta, \theta_0, \phi, P_s) + \frac{RT(\lambda, \theta, \theta_0, P_s)}{1 - RS_b(\lambda, P_s)}, \tag{1}$$

where $\lambda$ is wavelength, $\theta$ is the viewing zenith angle (VZA), $\theta_0$ the solar zenith angle (SZA), $\phi$ the relative azimuth angle (RAA), $P_s$ is the surface pressure, $I_0$ is the path scattering radiance by the atmosphere, calculated as the TOA radiance for a black surface, $T$ is total (direct+diffuse) solar irradiance reaching surface and reflected back the satellite multiplied by the transmittance, $S_b$ is the diffuse flux reflectivity of the atmosphere, and R is LER.

In order to calculate GLER, we use VLIDORT to simulate $I_{comp}$ for a clear sky over a non-Lambertian surface with the water leaving radiance model described in Sect. 2.2 along with the Cox-Munk slope distribution for surface roughness. A look-up table (LUT) approach is used to calculate TOA radiance operationally, as running VLIDORT can be computationally expensive. Details on the LUT used are available in Appendix B of Qin et al. (2019). Given TOA Radiance from VLIDORT,

we can then calculate GLER using Eqn. 1.



## 2.6 OMI Data and Selection Criteria

The measured LER data used to evaluate ocean GLER in this study were retrieved from OMI collection 3 level 1b Vis channel radiance data by inverting Eqn. 1 where $I_{obs}$ is used in place of $I_{comp}$. The OMI radiances are normalized using the OMI day-1 solar irradiance spectrum adjusted for variation in Earth-Sun distance when radiance measurements were collected. The GLER product is
designed to characterize the magnitude and the angular variability of the Earth's surface reflectance in a Rayleigh atmosphere, and therefore several aspects of instrument calibration must be considered. Absolute radiometric calibration error will introduce bias and inconsistency across the measurement swath for LER at any single wavelength.Dobber et al. (2008) estimated that the uncertainty
in radiometric calibration of OMI collection 3 sun-normalized radiances is under 2% and that the relative viewing angle dependence is also less than 2%. Schenkeveld et al. (2017) evaluated long-term changes in the absolute radiometric response of the OMI instrument and estimated degradation of approximately 1-1.5% over the lifetime of the mission in the wavelength region used in this study. Since we compare results at 354, 388, 440, and 466 nm in this study, the spectral dependence of
OMI calibration is also an important consideration. Little work has been published on this topic, although the study by Jaross and Warner (2008) compared the OMI sun-normalized radiances to radiative transfer model simulations over Antarctic ice, and showed that the spectral dependence of the OMI calibration is within the uncertainty of the absolute radiometric uncertainty.

Absorption by $O_2$-$O_2$ and $O_3$ were accounted for at 440 nm and 466 nm, but neglected for 354 nm
and 388 nm. Since GLERs were simulated for a Rayleigh-only atmosphere, pixels with absorbing aerosols are removed using the OMAERUV absorbing aerosol index AI ($|AI| > 0.5$ are removed) (Torres et al., 2007). We compare the evaluation for two independent cloud screening methods to determine which will better represent the GLER evaluation. The MODIS geometrical cloud fraction (GCF) is retrieved from the 15 $\mu$m $CO_2$ absorption region (Menzel et al., 2007) and is colocated to
the OMI FOV in the OMMYDCLD product (Joiner, 2014). The OMI Raman cloud product contains an ECF and cloud pressure based on rotational-Raman scattering in the UV wavelengths using the Cox-Munk distribution to model ocean surface reflectivity (Vasilkov et al., 2008).

We compare cases with and without sun glint separately because the reflection of light in each case is quite different. For comparisons excluding sun glint scenes, we screen out data with a co-
scattering angle of less than 20° in which sun glint can occur. For the comparisons with sun glint, the data are identified by evaluating the difference between OMI measured LER at 354 nm and 388 nm. The difference in LER occurs because of a spectrally dependent error in the underestimation of the Rayleigh scattering of diffuse light when one assumes a Lambertian ocean surface, when the reflectance is in fact specular. We select sun glint scenes when the difference between the mea-
sured LER at 354 nm and 388 nm is less than -0.05. This method does not require additional cloud screening, as the spectral dependence of clouds is quite small.





In addition to the OMI-derived LER, we compare with the Kleipool LER climatology (Kleipool et al., 2008), since a number of current operational algorithms use this LER data as input. There are two LER datasets available from the Kleipool data, one representing the monthly minimum LER and another determined through interpretation of LER histograms. Both are shown in our evaluation
as each is used in some algorithms.

## 3   Results and Discussion

### 3.1   Global Comparison of GLER and OMI-derived LER

Table 1: Statistical analysis of GLER vs OMI LER for non-sun glint scenes
January 2006 deep ocean only with raman ECF = 0.0 (number of points = 341,629)

| Wavelength | Slope | $R^2$ | Mean Bias | RMSE |
|---|---|---|---|---|
| 354 nm | 0.61 | 0.54 | -0.01 | 0.015 |
| 388 nm | 0.80 | 0.74 | 0.002 | 0.008 |
| 440 nm | 0.76 | 0.69 | 0.012 | 0.015 |
| 466 nm | 0.76 | 0.66 | 0.013 | 0.016 |

Table 2: Statistical analysis of GLER vs OMI LER for sun glint only scenes
January 2006 deep ocean only (number of points = 4,344)

| Wavelength | Slope | $R^2$ | Mean Bias | RMSE |
|---|---|---|---|---|
| 354 nm | 0.70 | 0.55 | -0.01 | 0.055 |
| 388 nm | 0.74 | 0.56 | -0.006 | 0.068 |
| 440 nm | 0.75 | 0.52 | 0.013 | 0.098 |
| 466 nm | 0.77 | 0.53 | 0.011 | 0.010 |

First we compare GLER with the OMI-derived LER globally for January 2006 at 4 wavelengths in Figs. 1 and 2. To determine a cloud screening method for the evaluation of sun glint free data, in
Fig. 1 we compare GLER with the OMI-derived LER using the cloud screening methods introduced in Sect. 2.6. We note there is a spectral dependence in the difference between GLER and the OMI-derived LER. At 354 nm GLER is biased high compared to the OMI-derived LER, whereas no bias exists at 388 nm and at 440 nm and 446 nm GLER are biased low. As shown in Table 1, the GLER and the OMI-derived LER compare best at 388 nm where $R^2$ is 0.74 and the bias is 0.002. For Vis
wavelengths, there appears to be two distributions of data in the scatterplot, which could possibly be related to aerosols in the OMI measured data. These issues with aerosols will be further analyzed in Sect. 3.3.



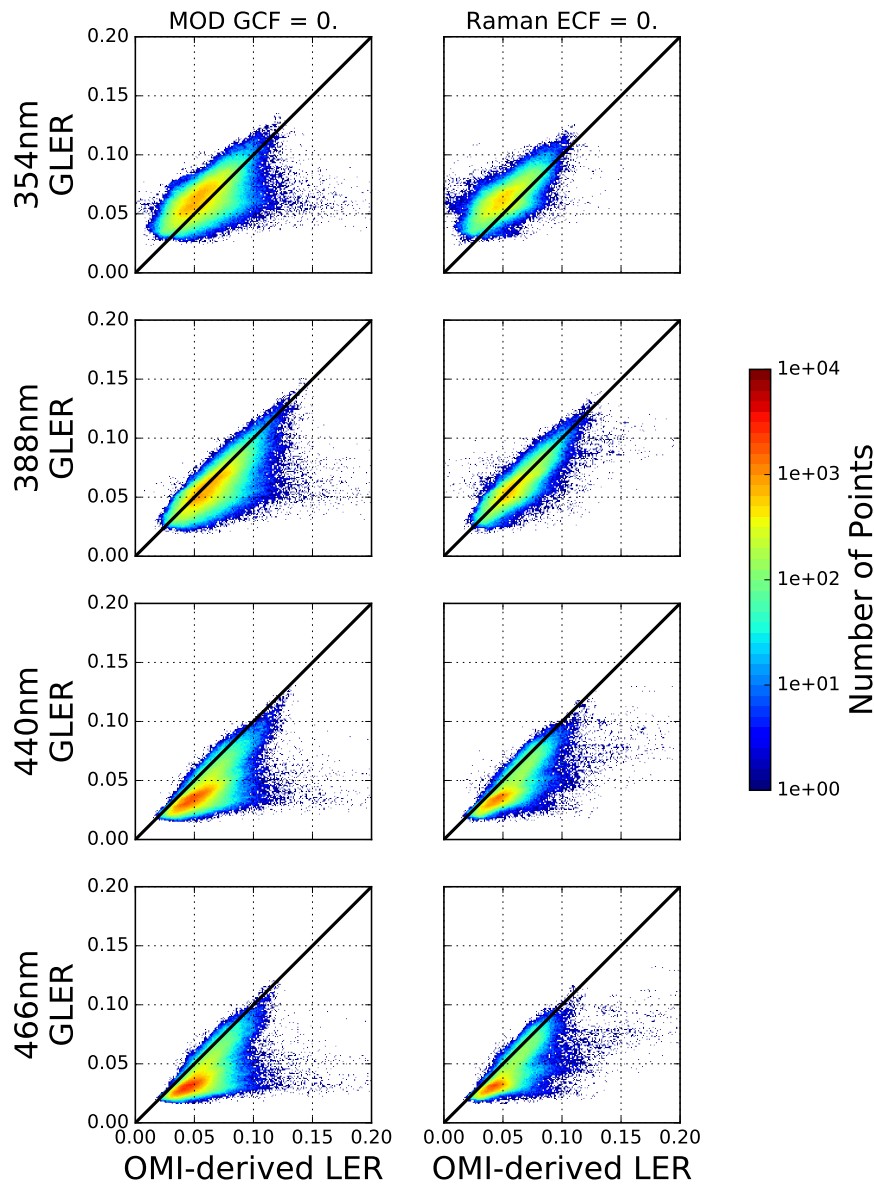

Fig. 1: Scatterplots of OMI-derived LER vs GLER for January 2006 with possible sun glint removed at 4 wavelengths (354 nm, 388 nm, 440 nm, and 466 nm). Data are for deep ocean (based on OMI level 1b ground pixel quality flags) and have been screened for aerosols (OMAERUV |AI| < 0.5). Clouds are screened through two different methods which are described in Sect. 2.6.



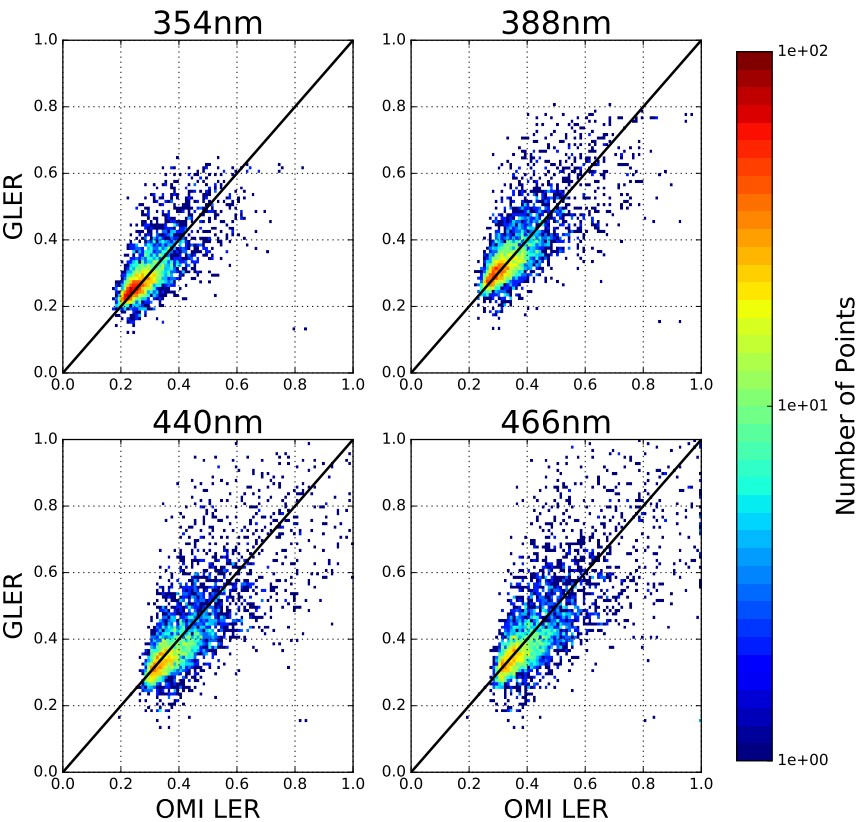

Fig. 2: Scatterplots of OMI-derived LER vs GLER for January 2006 for pixels with sun glint at 4 wavelengths (354 nm, 388 nm, 440 nm, and 466 nm). Data are for deep ocean with no screening for clouds or aerosols.

In general, the cloud screening methods produce similar results with only small differences that do not impact the overall evaluation. For the rest of the paper, the Raman ECF will be used for cloud screening as the Raman ECF shows better correlation than using the MODIS GCF which could simply be due to the fact that in 2006 there is a 15-minute overpass time difference between Aqua

5 and Aura. It is also worth noting since OMI has a wider swath than MODIS, MODIS cloud retrievals are not available for pixels on the edge of OMI swath (these pixels are not shown in Fig. 1).

In Fig. 2, the comparisons of GLER and OMI-derived LER are presented for data with sun glint. These data were not screened for cloud or aerosols, as strong glint can lead to artificial classification





of aerosol or clouds in the retrieval algorithms. It is evident that there are two main regions for the sun glint, a more general sun glint distribution with reflectivity between 0.2-0.4 and then a smaller distribution that exhibits extreme sun glint where LER reaches as high as 1. As shown in Table 2, the bias between GLER and the OMI-derived LER is smaller for the data with sun glint than the

sun glint-free data. For sun glint pixels, the bias is smaller at longer wavelengths where the water leaving radiance contributes the least. Despite the smaller bias, $R^2$ is worse for the sun glint cases at around 0.55 due to the increased sensitivity of GLER to the wind speed for sun glint scenes. For the brighter glint data, there is much more uncertainty and GLER is biased high compared with the OMI-derived LER. If the measured wind speed is too low, the model will overestimate the LER of

the glint, whereas when the measured wind speed is too high, the model will underestimate the LER of glint. This sensitivity to the wind speed will be further evaluated in Sect. 3.5. The small bias is possibly caused by aerosols which scatter or absorb the direct light causing a small dimming affect in the OMI glint data. This issue will be evaluated in Sect. 3.3.

### 3.2   Angular Behavior of GLER

Figures 3 and 4 show a comparison of the cross-track dependence of GLER and OMI-derived LER along with the Kleipool LER climatology for a few solar zenith angle ranges screened for clouds using the Raman ECF and screened for absorbing aerosols with the OMAERUV AI. GLER follows a similar cross-track pattern at various solar zenith angles as the OMI-derived LER that varies with wavelength. However, there is a bias between GLER and the OMI-derived LER which varies by

wavelength. For Vis wavelengths, the OMI-derived LERs are biased as much as 0.01-0.02 high compared to the GLER, whereas for UV wavelengths the bias is nearly zero at 388 nm and at 354 nm GLER is biased around 0.01 higher than the OMI measurements. In the UV channels, especially at 354 nm, the bias varies both with cross-track and solar zenith angle. Outside of the sun glint, where water leaving radiance dominates the reflectance due to increased diffuse illumination, reflectivity

decreases with increasing wavelength because pure water absorption increases with wavelength.

The Kleipool LER climatology compares well with the OMI-derived LER near nadir but does not capture any of the BRDF effects seen in both GLER and OMI-derived LER since it is a climatology that averages all viewing geometries. We note that there is a slight increase in Kleipool LER at higher view zenith angles, but this is simply due to the sampling used for this analysis. Compared

to the OMI-derived LER, the Kleipool data have a cross-track-dependent difference of as much as 0.04 outside of the sun glint and 0.1 in the sun glint for UV wavelengths. The difference between Kleipool and the OMI-derived LER (0.01) is smaller for Vis wavelengths outside of the sun glint. At the higher solar zenith angles, the Kleipool monthly data appear to be adversely affected either by ice or residual cloud that was not fully removed from the Kleipool LER climatology. This is

evident by the large bias compared with OMI-derived LER as well as the large variability coming

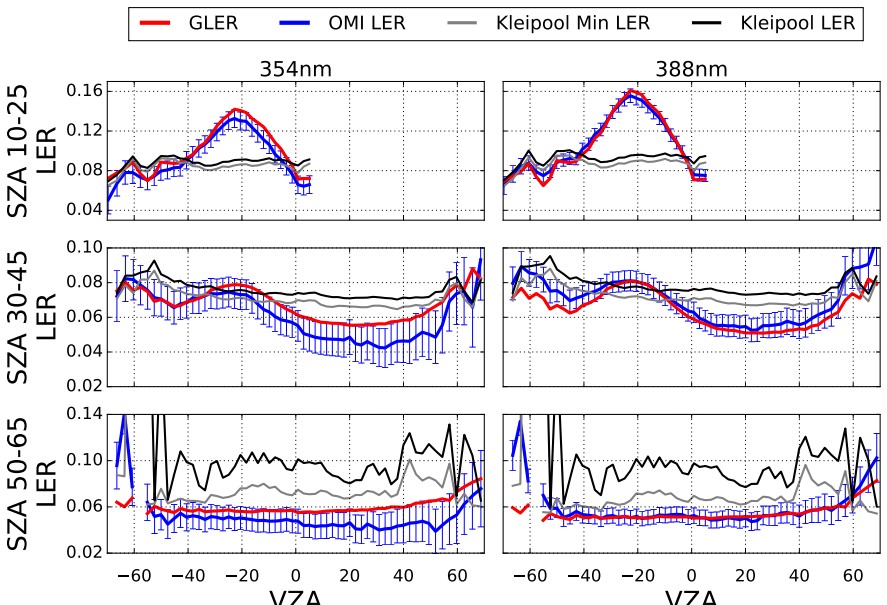

Fig. 3: January 2006 LER as a function of VZA for select SZA ranges at 354 nm and 388 nm over the Pacific Ocean (only deep ocean pixels considered). The blue error bars represent the 2% calibration uncertainty of OMI (Dobber et al., 2008). Negative VZAs represent the west side of the OMI swath (forward scattering), whereas positive VZAs represent the east side of the OMI swath (backward scattering). Data are screened for clouds with the Raman ECF and aerosols are removed with the OMAERUV AI.

from spatial sampling.

### 3.3 Simulating GLER with Aerosols

As noted in earlier sections, aerosols can have an impact on measured LER. For this reason, a simulation was performed to calculate GLER including the effect of aerosols. Aerosol-related input
5   parameters to VLIDORT (layer AOD, single scattering albedo [SSA], and phase function) are from MERRA-2 aerosol reanalysis data which is produced using the GEOS-5 atmospheric model and data assimilation system. MERRA-2 assimilates radiance data from a variety of satellite sensors which are then used to train a neural network to produce AOD which is calibrated to the Aerosol Robotic Network (AERONET) direct-Sun point measurements (Randles et al., 2017; Gelaro et al.,
10   2017). The species-specific aerosol scattering functions are characterized by six sets of generalized



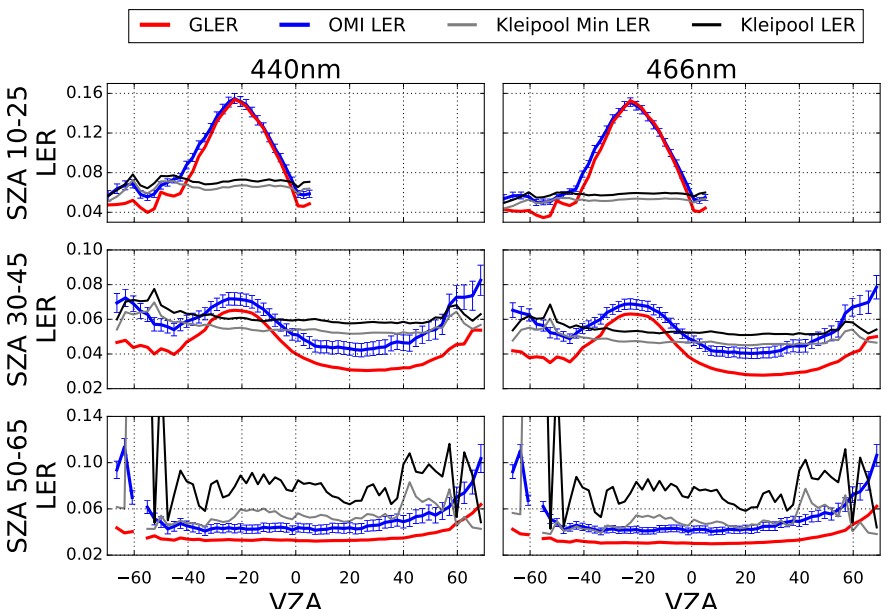

Fig. 4: Same as Fig. 3 but for 440 nm and 466 nm

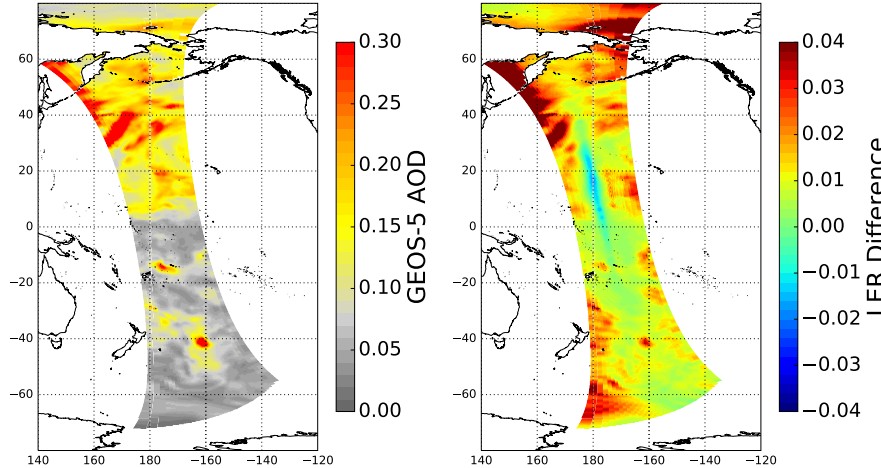

Fig. 5: Map showing the results from GLER aerosol simulation for April 10, 2006 Orbit 09229. On the left is GEOS-5 470 nm AOD used in the simulation and on the right is change in GLER when the aerosol contribution is added to the simulations at 466 nm.





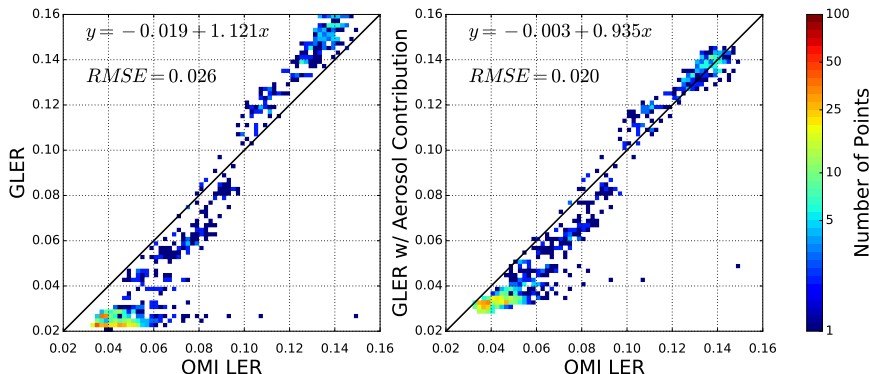

Fig. 6: Comparison of GLER versus OMI-derived LER for April 10, 2006 Orbit 09229. On the left is GLER compared to OMI-derived LER at 466 nm, whereas on the right is GLER with contribution from aerosols vs OMI-derived LER for 466 nm. All data are filtered to remove clouds with the Raman ECF

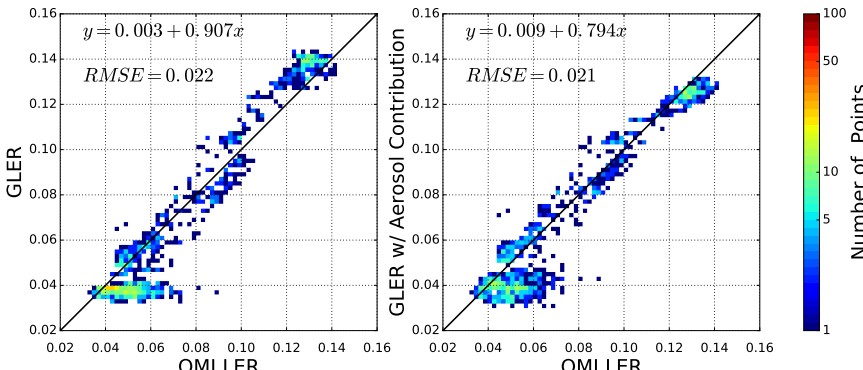

Fig. 7: Same as Fig. 6 but for 354 nm

spherical function expansion coefficients generated from Mie theory.

The simulation was performed for orbit 9229 on April 10, 2006 over the central Pacific in order to determine the aerosol effect for a general background oceanic aerosol case. Figure 5 shows that the AOD for this orbit ranged from around 0.05 in the South Pacific gyre to larger than 0.4 in the northern Pacific. The impact of the aerosols on GLER ranges from a decrease in LER of 0.01 in sun glint to an increase in LER of greater than 0.04 outside of the glint, especially at higher viewing angles. Figures 6 and 7 show the comparison between GLER and the OMI-derived LER as well as a comparison between GLER with a contribution from aerosols and the OMI-derived LER.

In Fig. 6 the addition of aerosols at 466 nm increases the LER by about 0.01 over darker surfaces,



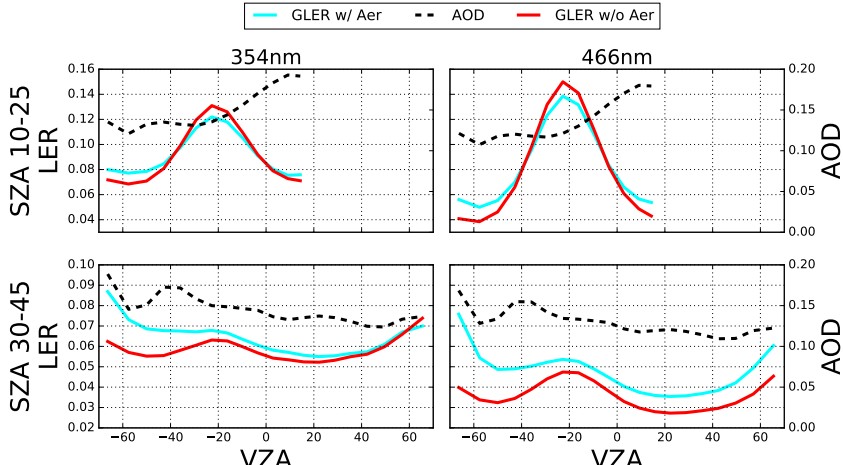

Fig. 8: LER as a function of VZA at 354 nm and 466 nm for April 10, 2006 Orbit 09229. Data are cloud screened with the Raman ECF, but have not been screened for aerosols. As in Fig. 3 negative VZAs represent the west side of the OMI swath whereas positive VZAs represent the east side of the OMI swath.

whereas the brighter regions which have some sun glint contribution show a reduction in LER of around 0.01. The combination of these changes improves the slope from 1.121 before considering aerosols to 0.935 after aerosols are introduced. There is also improvement of the root mean squared error (RMSE) which decreases from 0.026 to 0.02. After accounting for aerosols, OMI-derived LER

is still around 0.01 higher than GLER at 466 nm for darker surfaces. The brighter surfaces, however, which have some sun glint contribution, have little to no bias after accounting for aerosols.

At 354 nm the aerosol impact is not as significant as that seen for 466 nm with on the whole little to no change in the bias for darker surfaces and a reduction of around 0.01 for the brighter surfaces. In Fig. 8 there is a cross-track dependence in the aerosol contribution to GLER at 354 nm.

For geometries with forward scattering (negative VZAs), the aerosol contribution can effectively increase the derived LER by 0.01 or more, whereas for backward scattering geometries (positive VZA) there is little to no change. This view angle dependence of the aerosol impact would remove the crosstrack dependent bias seen in Fig. 3 resulting in GLER being approximately 0.01 higher than the OMI-derived LER at 354 nm for all view angles.

Overall we note that at 466 nm an AOD of 0.1-0.15 can increase LER by around 0.01 in the backscattering direction, while increasing LER by as much as 0.02 in the forward scattering direction. At 354 nm, however, similar AOD values have little impact in the back scattering direction, but can increase LER by as much as 0.01 in the forward scattering direction.


## 3.4 Inter-annual Variability of LER

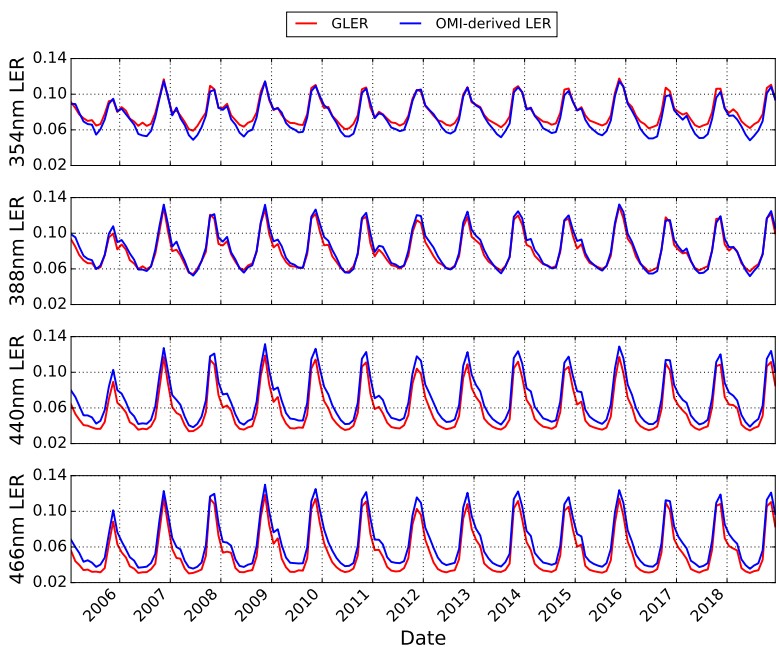

Fig. 9: GLER and OMI-derived LER in equatorial Pacific ocean (180W-120W, 30S-0) for 4 wavelengths (354, 388, 440, and 466 nm) as a function of time for the OMI mission from row 10. Data are filtered to remove clouds with the Raman ECF.

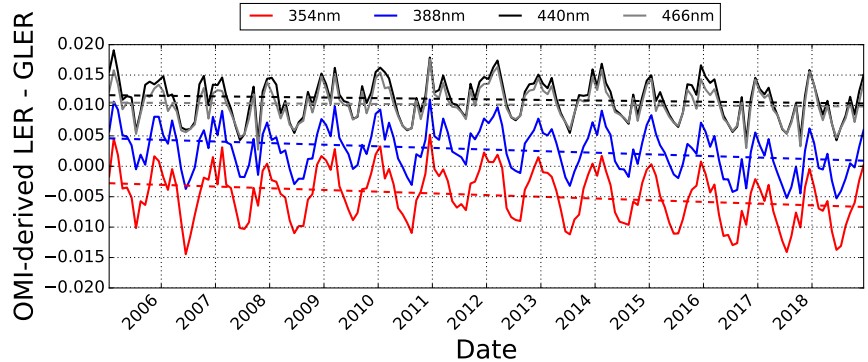

Fig. 10: Trend in the difference between GLER and the OMI-derived LER for the same location as in Fig. 9 at UV and Vis wavelengths for row 10. Data are filtered to remove clouds with the Raman ECF.





Surface LER over the ocean can change day to day depending on the chlorophyll concentration which affects the water leaving radiance contribution as well as due to changes in the wind speed which can alter the roughness of the water surface. There is also a seasonal variation in LER due to the changing viewing geometry of satellite measurements as the SZA changes through the year.

In Figs. 9 and 10, GLER and OMI-derived LER are shown for the duration of the OMI mission to evaluate the ability of GLER to capture these variations. In this figure, data were selected for a region in the south Pacific gyre (180W-120W, 30S-Equator) as this region is generally free of strong aerosols and has relatively low cloud fractions. OMI row 10 is evaluated to avoid sun glint as well as to avoid the OMI row anomaly which impacts many of the OMI rows starting around 2009 (Levelt

et al., 2018). While this relatively small bias between GLER and the OMI-derived LER is evident, GLER does follow the same general trend as the OMI measured data.

Figure 10 shows there is about a 0.01 seasonal variation in the GLER and OMI-derived LER difference for all wavelengths which could possibly be due to seasonal changes in aerosols or seasonality in cirrus clouds that are too thin to be retrieved by the Raman ECF. We note that in Fig. 10

there is a small downward trend in the difference between GLER and OMI-derived LER of at most 0.005 LER. This may be related to the downward drift in the OMI measurements which is known to be about 2% in TOA radiance over the duration of the mission.

### 3.5 Sensitivity to Chlorophyll and Wind Speed

Table 3: Sensitivity to wind speed

| Wavelength | Mean Abs Diff | Max Abs Diff |
|------------|---------------|--------------|
| 354 nm | 0.0010 | 0.16 |
| 388 nm | 0.0013 | 0.21 |
| 440 nm | 0.0015 | 0.27 |
| 466 nm | 0.0016 | 0.30 |

Table 4: Sensitivity to chlorophyll

| Wavelength | Mean Abs Diff | Max Abs Diff |
|------------|---------------|--------------|
| 354 nm | 0.0043 | 0.0067 |
| 388 nm | 0.0047 | 0.0087 |
| 440 nm | 0.0022 | 0.0047 |
| 466 nm | 0.0012 | 0.0019 |

In order to determine the uncertainty in GLER calculations a sensitivity test was performed based

on the inputs of chlorophyll and wind speed. The wind speed measurements were perturbed +/-





Table 5: Sensitivity to wind speed and chlorophyll

| Wavelength | Mean Abs Diff | Max Abs Diff |
|---|---|---|
| 354 nm | 0.0044 | 0.17 |
| 388 nm | 0.0049 | 0.21 |
| 440 nm | 0.0028 | 0.27 |
| 466 nm | 0.0021 | 0.30 |

$1\,\mathrm{m\,s^{-1}}$ as Wentz and Meissner (2000) note that is the uncertainty in their wind speed algorithm. The MODIS Ocean Color Team notes that the chlorophyll uncertainty varies regionally but can possibly be as high as 35% (Moore et al., 2009). We perturb chlorophyll by this 35% in order to gather an absolute bound of GLER to the chlorophyll input. We place an absolute lower bound on the wind

speed in our calculations of $0.4\,\mathrm{m\,s^{-1}}$ and a lower bound of 0.01 of $\mathrm{mg\,m^{-3}}$ on the chlorophyll data sets as measurements of these input below these lower bounds are unrealistic and could lead to large errors in calculation of GLER. In Tables 3, 4, and 5 the magnitude of the mean and max differences are reported in units of LER.

As seen in Table 3 the average wind speed sensitivity is quite small at around 0.0015, but the

maximum sensitivity can become as high as 0.3 due to the high sensitivity of sun glint to wind speed. This is because for extreme sun glint cases a small roughness in the ocean surface will lead to increased scattering of light which will significantly diminish the strength of the glint. It is worth noting that while such roughness decreases the strength of the glint, it will increase the overall size of the region affected by sun glint due to the scattering of light at the ocean surface. The wind speed

sensitivity decreases with decreasing wavelength because the fraction of the direct solar light which is responsible for sun glint decreases for shorter wavelengths where the contribution of the diffuse light increases due to Rayleigh scattering.

In Table 4 the chlorophyll sensitivity is shown to be much smaller than the wind speed sensitivity. In contrast with the wind speed sensitivity, the chlorophyll sensitivity is largest at UV wavelengths

because the CDOM absorption, which rises with increasing chlorophyll according to the Case 1 water model, exponentially increases for shorter wavelengths.

When both wind speed and chlorophyll sensitivity are combined, Table 5 shows the mean difference in GLER is similar to that obtained with just the chlorophyll sensitivity. This is likely because while the wind speed has a significant impact on sun glint, only a small fraction of OMI pixels are

impacted by glint.



### 3.6 Additional Sources of Uncertainty

We note that in addition to the sensitivities from the input data, uncertainties also result from the modeling of the GLER as well as the OMI data used for the evaluation. One possible source of uncertainty is the water leaving radiance model being used in the calculations. Here we implement a

Case 1 water model that is assumed to be representative of the global oceans with dependence only on chlorophyll. Szeto et al. (2011), however, showed that the world's oceans are optically variable and that optical parameters such as chlorophyll absorption vary even for Case 1 open ocean. Work by Lee et al. (2006) similarly showed that large deviations exist from the presumed Case 1 water model due to uncertainty in the optical properties used to parameterize the models. They determined

that the SeaWiFS remote sensing reflectance retrievals at 555 nm for Case 1 water have a deviation of +/- 50% from that of a Case 1 bio-optical property model.

Our simulations do not include any vibrational-Raman scattering effects which can increase the water leaving radiance. Westberry et al. (2013) show that Raman scattering can impact the water leaving radiance in Vis wavelengths as much as 4-7% for low chlorophyll concentrations.

In Sect. 3.3 is it shown that aerosols can increase the LER derived from OMI. Uncertainty in AOD used for this analysis could have an appreciable impact on the evaluation results. Randles et al. (2017) compare the MERRA-2 550 nm AOD with the Maritime Aerosol Network (MAN) and find the MERRA-2 AOD to be biased low by 0.009 with large spread in the comparison which they note could be due to the uncertainty of the MAN AOD of +/- 0.02. Even though the AOD

uncertainties are small, we have shown that even a 0.05-0.1 AOD increase can increase 466 nm LER by 0.005-0.01.

With regard to the OMI measurements, uncertainty could arise from the cloud screening of the OMI measurements as retrieval of cloud properties can become difficult for thin cirrus clouds, which are especially prevalent over the western Pacific (Nazaryan et al., 2008). It has been shown that such

contamination can actually increase MODIS AOD retrievals by 0.015-0.025 (Kaufman et al., 2005). As previously mentioned, there is also up to a 2% TOA radiance absolute calibration uncertainty with the OMI measurements (Dobber et al., 2008) which can lead to an uncertainty of around 0.01-0.02 in LER in the UV and up to 0.005 LER in the Vis. Several OMI algorithm teams apply adjustments to remove residual error in viewing angle dependence in the OMI measurements by looking at Earth

radiances over land. When these corrections were applied in our evaluation of GLER, we found the difference between GLER and the OMI measurements in the UV and Vis was reduced.

Given the uncertainties listed above, we believe differences between GLER and OMI measurements are likely caused by some combination of these factors. It is possible that these factors vary with wavelength. For example, we have shown that aerosols have a greater impact at 440 nm and

466 nm, whereas other factors such as chlorophyll uncertainty are more important at 354 nm and



388 nm where the water leaving radiance contributes more than the direct reflectance.

## 4 Conclusion

Previous work Qin et al. (2019) introduced the GLER product for land surfaces based on BRDF input from MODIS. In this paper we evaluate a surface LER product called GLER which accounts

for the ocean surface BRDF effects at UV and Vis wavelengths. Surface roughness is modeled using the Cox-Munk slope that depends on wind speed measurements from the AMSR-E and SSMIS instruments. A contribution of water leaving radiance is also included which is based on chlorophyll input from MODIS.

We evaluated the GLER product over water by comparing with OMI-derived LER at UV and Vis

wavelengths. The BRDF effect in the OMI-derived LER is captured well with the GLER data at Vis wavelengths and small view angle dependence in the difference for UV wavelengths which could possibly be explained by the anisotropic scatter of aerosols. There is, however, a bias between GLER and the OMI-derived LER which is less than 0.01 LER after accounting for the effect of aerosols.

We note the GLER data capture the seasonality and inter-annual variability seen in the OMI mea-

surements that may be caused by variations in the viewing angles as well as changes in the chlorophyll and wind speed data due to meteorological phenomena. The bias between GLER and OMI-derived LER could be caused by a combination of things including small calibration errors in the OMI-derived LER, deviations in the OMI measurements from the Case 1 water model, and residual thin cloud that are difficult to screen.

There are several possible applications for the GLER product. It can be used to replace climatological LER's currently used by many cloud, trace gas, and aerosol algorithms. Additionally, GLER can be used as a tool to evaluate satellite calibration to detect possible drift or striping in instrument retrievals. In future missions such as PACE, GLER can be adapted to perform retrievals of water leaving radiance at UV and Vis wavelengths.

In future work we plan to implement a Case 2 water model for turbid and coastal waters as well as replace our LUT approach with a neural network approach to reduce the computational time to produce the GLER product.

## Appendix A  Description of the VSLEAVE Water-Leaving Radiance Model

In this Appendix, we give details of the water-leaving radiance scheme included in the VSLEAVE

supplement to VLIDORT Version 2.8. Section A1 of this appendix deals with the basic water leaving formulation, while Sect. A2 deals with the ocean optics model. In particular, the material in Sect. A2





is based on the work of Sayer et al. (2010), which has a comprehensive review of semi-empirical marine optics formula, and a companion paper Sayer et al. (2017), the latter containing important updates to the optics model. The treatment is for Case 1 waters.

**A1    Water-Leaving Radiance Model**

Here we summarize the computation of water-leaving radiances using the VSLEAVE supplement to VLIDORT. A full description of the VLIDORT VSLEAVE supplement used here may be found in Spurr et al. (2019). Water-leaving output from VSLEAVE consists of three terms which are sun-normalized radiances. The first is a direct term $L_{w,direct}(\mu, \mu_o, \phi)$ which is the water leaving radiance for solar illumination angle $\theta_0$ and cosine $\mu_0 = \cos(\theta_o)$ going into the viewing direction

with zenith angle $\theta$ and $\mu = \cos\theta$ and the relative azimuth angle $\phi$ between the solar and viewing directions.

The other two water-leaving radiance outputs may be written $L_{w,m}(\mu, \mu_0)$ and $L_{w,m}(\mu_i, \mu_0)$, where $\mu_i (i = 1, \ldots N_d)$ are the discrete-ordinate polar cosines, and $m$ is the Fourier component index, $m = 0, 1, \ldots 2N_d - 1$. These are diffuse-term contributions: $L_{w,m}(\mu_i, \mu_0)$ is required for

the inclusion of surface leaving in the diffuse-scattering boundary condition at surface, while the term $L_{w,m}(\mu, \mu_0)$ is required for post-processing of the discrete ordinate solution. Fourier terms arise from cosine-azimuth expansions of the full functions: $L_{w,direct}(\mu, \mu_0, \phi) = L_{w,0}(\mu, \mu_0) + 2\sum_{m=1}^{\infty} L_{w,m}(\mu, \mu_0) \cos(m(\phi))$. In the discrete-ordinate approximation with $N_d$ streams, we can only use $2N_d - 1$ components in this sum. In the post-processing, it is more accurate to use the

complete term $L_{w,direct}(\mu, \mu_0, \phi)$ itself in place of the (less-accurate) Fourier-series truncation, and this "exact-term correction" is the default in VSLEAVE. In this case, Fourier terms $L_{w,m}(\mu, \mu_0)$ are not needed. Note that we will always need the Fourier components $L_{w,m}(\mu_i, \mu_0)$ for the diffuse-field calculation. However, when there is no azimuth dependence, only $L_{w,0}(\mu_i, \mu_0)$ for $m = 0$ survives. In this study we consider an anisotropic distribution of the water-leaving contri-

bution, but the model can also generate it as an isotropic term $L_{w,iso}(\mu_0)$ which depends only on the incoming solar direction (no azimuth dependence, all outgoing directions equal), in which case $L_{w,m}(\mu, \mu_0) = 0 (m \geq 1)$ and $L_{w,0}(\mu, \mu_0) = L_{w,iso}(\mu_0)$ for all outgoing polar directions $\mu$, and also $L_{w,direct}(\mu, \mu_0, \phi) = L_{w,iso}(\mu_0)$. The current default for VSLEAVE is for an unpolarized azimuth-independent formalism. Thus only the intensity component of the water-leaving Stokes

vector is non-zero, and there is no azimuthal dependence.

Water-leaving radiance may be written as

$$L_w(\mu, \mu_0, \phi) = L_w^*(\mu, \mu_0, \phi; \mathrm{Chl}, \mathrm{V}) \, T_{atm}(\mu_0), \tag{A1}$$

for any given combination of angles, where the transmittance $T_{atm}(\mu_0)$ depends only on the solar angle, and $L_w^*(\mu, \mu_0, \phi, \mathrm{Chl}, \mathrm{V})$ is computed from the marine optical properties using a semi-





empirical model which depends explicitly on the chlorophyll concentration and wind speed $V$. The ocean-optics model for the determination of $L_w^*$ is described below.

### A2  Ocean-Optics Model

Though the longest wavelength in the GLER product is presently 466 nm, we describe our model of
ocean optics over a wider spectral range below as the model has applications in aerosol and ocean color studies where the longer visible and near-infrared wavelengths are typically used. The water absorption $\alpha_W(\lambda)$ coefficients have been linearly interpolated from a table of values at every 5 nm from 200-900 nm constructed from a number of literature sources. These are Quickenden and Irvin (1980), interpolated with Lee et al. (2015) between 325 and 345 nm; (2) 350-550 nm from Lee et
al. (2015); Pope and Fry (1997) for 555-725 nm; Hale and Querry (1973), table 1, for 725-900 nm (the latter with 25 nm increments linearly interpolated to 5 nm values]. Table entries provided as extinction coefficients $k_W$ are converted using $\alpha_W(\lambda) = \frac{4\pi k_W}{\lambda}$, where wavelengths are in m and extinctions in m$^{-1}$.

The chlorophyll absorption $\alpha_{ph}(\lambda)$ comes from two sources. The first source (in the range 300-
400 nm) relies on linear interpolation of two sets of coefficients $\{a_1(\lambda), b_1(\lambda)\}$ given at 10 nm intervals in this range (Vasilkov et al., 2005). The absorption is given by

$$\alpha_{Ph}(\lambda, \text{Chl}) = \text{Chl} \cdot a_1(\lambda) \cdot \text{Chl}^{-b_1(\lambda)}, \tag{A2}$$

where Chl is the chlorophyll concentration and $\lambda$ is . The value at 300 nm is used for all $\lambda < 300$ nm. The second source (over the range 400-720 nm) is based on linear interpolation of two sets of
coefficients $a_2(\lambda), b_2(\lambda)$ at 10 nm intervals (Lee et al., 2005). The absorption formula in this regime is given by

$$\alpha_{Ph}(\lambda, \text{Chl}) = [a_2(\lambda) + b_2(\lambda) \ln(a_{440})] a_{440}, \tag{A3}$$

where $a_{440} = 0.06 \cdot \text{Chl}^{0.65}$ (Morel and Maritorena, 2001). The value at 720 nm is used for all $\lambda > 720$ nm. The CDOM absorption is given by Morel and Maritorena (2001):

$$\alpha_{CDOM}(\lambda, \text{Chl}) = 0.2 \cdot \left(\alpha_w(440\,\text{nm}) + 0.06 \cdot \text{Chl}^{0.65}\right) exp\left[-0.014(\lambda - 440)\right], \tag{A4}$$

where $\alpha_w(440\,\text{nm}) = 0.00635$ and $\lambda$ is in nm. The complete absorption is then

$$\alpha_{TOT}(\lambda, \text{Chl}) = \alpha_W(\lambda) + \alpha_{Ph}(\lambda, \text{Chl}) + \alpha_{CDOM}(\lambda, \text{Chl}). \tag{A5}$$

We use the following formula for the backscattering coefficient, assuming it is half of the scattering coefficient for pure water Rayleigh scattering, as described in Morel et al. (2007):

$$b_W(\lambda) = 0.0028 \left(\frac{420}{\lambda}\right)^{4.3}, \tag{A6}$$



with $\lambda$ in nm. This For the particulate matter backscattering coefficient, we use the following from Morel and Maritorena (2001):

$$b_{Ph}\left(\lambda, \mathrm{Chl}\right) = b_{Pb}\left(\mathrm{Chl}\right)\beta_{bbp}\left(\mathrm{Chl}, \lambda\right) \tag{A7}$$

$$b_{Pb}\left(\mathrm{Chl}\right) = 0.416C^{0.766}; \quad \beta_{bbp}\left(\mathrm{Chl}, \lambda\right) = 0.002 + 0.01\left[0.5 - 0.25log_{10}\mathrm{Chl}\right]\left(\frac{\lambda}{550}\right)^{V}, \tag{A8}$$

where the exponent $V = 0$ for $\mathrm{Chl} > 2$ , and $V = 0.5\left[log_{10}\mathrm{Chl} - 0.3\right]$ for $\mathrm{Chl} \leq 2$. The complete backscattering is then

$$b_{TOT}\left(\lambda, \mathrm{Chl}\right) = b_{W}\left(\lambda\right) + b_{ph}\left(\lambda, \mathrm{Chl}\right). \tag{A9}$$

In the original formulation of water-leaving radiance in VLIDORT, the following formula was used to obtain the basic ocean-surface reflectance (Morel and Gentili, 1992):

$$R\left(\mathrm{Chl}, \lambda, \mu_0\right) = f\left(\mu_0\right)R_{TOT}\left(\lambda, \mathrm{Chl}\right) \equiv f\left(\mu_0\right)\frac{b_{TOT}\left(\lambda, \mathrm{Chl}\right)}{a_{TOT}\left(\lambda, \mathrm{Chl}\right)} \tag{A10}$$

$$f\left(\lambda, \mathrm{Chl}, \theta_0\right) = d_0 - d_1\eta - d_2\eta^2 + \left(d_3\eta - d_4\right)\mu_0; \quad \eta = \frac{b_W\left(\lambda\right)}{b_{TOT}\left(\lambda, \mathrm{Chl}\right)}. \tag{A11}$$

Here, $f(\lambda, \mathrm{Chl}, \theta_0)$ is given with 5 constants $\{ d_0, d_1, d_2, d_3, d_4 \} = \{ 0.6279, 0.0227, 0.0513, 0.2465, 0.3119 \}$ , and $\mu_0 = \cos(\theta_0)$ is the cosine of the solar zenith angle. In order to assign the water-leaving radiance, the complete reflectance term is given by

$$R'\left(\mathrm{Chl}, \lambda, \mu_0\right) = \frac{R\left(\mathrm{Chl}, \lambda, \mu_0\right)}{1 - \omega R\left(\mathrm{Chl}, \lambda, \mu_0\right)}. \tag{A12}$$

Here, albedo $\omega = 0.485$, using the value in Austin (1974). The isotropic water-leaving radiance is then obtained after passage through the air-ocean interface:

$$S_{iso}\left(\mathrm{Chl}, \lambda, \mu_0\right) \approx \frac{\mu_0}{\pi}T_{Surf}\left(\theta_0\right)\frac{R'\left(\mathrm{Chl}, \lambda, \mu_0\right)}{\left|n_w\right|^2}. \tag{A13}$$

Here, $n_w$ is the relative refractive index of water to air. For the flat surface case, the air-water
boundary transmittance $T_{Surf}\left(\theta_0\right)$ is often set to 1.0. In practice we use Fresnel optics to compute this quantity; values are typically 0.96 or more, depending on the value of $\theta_0$. In the rough surface case, $T_{Surf}\left(\theta_0\right)$ may be computed using glitter calculations based on Gaussian probability wave-facet distributions characterized by wind-speed and direction.

The above formulation does not account for the atmospheric transmitted flux $T_{atm}\left(\theta_0\right)$ at the
ocean surface, a quantity which is propagated through the interface. In the previous formulation, the ratio $\frac{T_{atm}(\theta_0)}{Q}$ was made implicit in the factor $\frac{\mu_0}{\pi}$ appearing in Eqn. A13. Also, we replace





the $f(\lambda, \text{Chl}, \theta_0)$ calculation with the direction-dependent ratio $\rho \equiv f/Q$ from Morel and Gentili (1996); Morel et al. (2002). The water-leaving radiance is then:

$$S(\text{Chl}, \lambda, \theta_0, \mu, \varphi) = \mu_0 T_{atm}(\theta_0) T_{Surf}(\theta_0) \frac{R^*(\text{Chl}, \lambda, \theta_0, \mu, \varphi)}{|n_w|^2} \tag{A14}$$

$$R^*(\text{Chl}, \lambda, \theta_0, \mu, \varphi) = \frac{\rho(\text{Chl}, \lambda, \theta_0, \mu, \varphi) R_{TOT}(\lambda, \text{Chl})}{1 - \omega f(\text{Chl}, \lambda, \theta_0) R_{TOT}(\lambda, \text{Chl})}, \tag{A15}$$

where $R_{TOT}(\lambda, \text{Chl})$ is as defined in Eqn. A10, and $\rho$ is the ratio $f/Q$. We use a tabulated form of the ratio $f/Q$ in our calculations.

In order to obtain an isotropic surface leaving radiance, we derive a quantity $\bar{\rho}(\text{Chl}, \lambda, \theta_0)$ from the $f/Q$ tables by averaging over all outgoing zenith and relative azimuth angles, $\theta$ and $\phi$ , then interpolating linearly with wavelength $\lambda$ , followed by cubic spline interpolation and linear inter-

polation with the solar angle cosine $\mu_0$ and with the logarithm of the chlorophyll concentration. Spline interpolation is necessary because we want smooth and continuous derivatives with respect to Chl when considering linearization, as discussed below. The quantity $\bar{\rho}(\text{Chl}, \lambda, \theta_0)$ then defines the isotropic water-leaving contribution through:

$$S_{iso}(\text{Chl}, \lambda, \theta_0) = \mu_0 T_{atm}(\theta_0) T_{Surf}(\theta_0) \frac{\bar{R}^*(\text{Chl}, \lambda, \theta_0)}{|n_w|^2} \tag{A16}$$

$$\bar{R}^*(\text{Chl}, \lambda, \theta_0) = \frac{\bar{\rho}(\text{Chl}, \lambda, \theta_0) R_{TOT}(\lambda, \text{Chl})}{1 - \omega f(\text{Chl}, \lambda, \theta_0) R_{TOT}(\lambda, \text{Chl})}. \tag{A17}$$

The azimuth dependence is very weak in the $f/Q$ tables, and we have omitted this dependence in the surface leaving formulation. However, we can derive non-isotropic surface-leaving $f/Q$ values by interpolating table entries with the cosine of the outgoing angle $\mu$. The resulting table extractions are then $\tilde{\rho_v}(\text{Chl}, \lambda, \theta_0, \mu_v)$ and $\tilde{\rho_d}(\text{Chl}, \lambda, \theta_0, \mu_d)$ for each viewing angle $\mu_v$ and discrete ordinate

stream $\mu_d$; these quantities are azimuth-averaged. We then have

$$S_v(\text{Chl}, \lambda, \theta_0, \mu_v) = \mu_0 T_{atm}(\theta_0) T_{Surf}(\theta_0) \frac{R_v^*(\text{Chl}, \lambda, \theta_0, \mu_v)}{|n_w|^2} \tag{A18}$$

$$R_v^*(\text{Chl}, \lambda, \theta_0, \mu_v) = \frac{\tilde{\rho_v}(\text{Chl}, \lambda, \theta_0, \mu_v) R_{TOT}(\lambda, \text{Chl})}{1 - \omega f(\text{Chl}, \lambda, \theta_0) R_{TOT}(\lambda, \text{Chl})}, \tag{A19}$$

and similarly for the discrete ordinate directions.

In the rough-surface case, the above analysis for the ocean reflectance still holds, but now we

need to generate glitter-dependent transmission terms through the water-air interface, both for the incoming solar directions $\overrightarrow{T}_{aw}(\theta_0)$, and for outgoing line-of-sight $\overleftarrow{T}_{wa}(\theta_0, \mu_v)$ and discrete-ordinate $\overleftarrow{T}_{wa}(\theta_0, \mu_d)$ directions respectively. Thus for instance, the rough surface water-leaving term for a viewing angle $\mu_v$ is

$$S_{v,RS}(\text{Chl}, \lambda, \theta_0, \mu_v) = \mu_0 T_{atm}(\theta_0) \overrightarrow{T}_{aw}(\theta_0) \frac{R_v^*(\text{Chl}, \lambda, \theta_0, \mu_v)}{|n_w|^2} \overleftarrow{T}_{wa}(\theta_0, \mu_v), \tag{A20}$$





by analogy with Eqn. A18 and using Eqn. A19.

### Appendix B  Coupling of VLIDORT and VSLEAVE

The simplest approximation to $T_{atm}(\mu_0)$ is the *decoupled* scenario where the transmittance has no dependence on ocean properties. In this case, we drop the $T_{atm}(\mu_0)$ term from the main VSLEAVE

result in Eqn. A1 above, and then re-introduce $T_{atm}(\mu_0)$ from an internal computation in the main VLIDORT model. The direct transmittance $T_{direct}(\mu_0) = \exp[-\tau_{atm}/\mu_0]$ where $\tau_{atm}$ is the total atmospheric vertical optical depth; a closer value which includes a diffuse transmittance component is

$$T_{atm}(\mu_0) = exp\left[-\frac{1}{2}\frac{\tau_{atm}}{\mu_0}\right]. \tag{B1}$$

This equation was adapted from a similar formula in Gordon and Wang (1994). Equation B1 is easy to implement in VLIDORT. A more accurate expression may be obtained in certain cases by using a pre-calculated look-up table of $T_{atm}(\mu_0)$ values, computed offline with VLIDORT in a Rayleigh atmosphere over a 270-900 nm wavelength range, and for a number of $\theta_0$. However, $T_{atm}(\mu_0)$ is still decoupled from the VSLEAVE water-leaving radiance output.

The coupling scheme works as follows. From Eqn. A1, we write

$$L_w(\mu, \mu_0) = L_w^*(\mu, \mu_0)\,T^\downarrow(\mu_0), \tag{B2}$$

where $T^\downarrow(\mu_0)$ is the total (direct and diffuse) downwelling atmospheric transmittance at the ocean surface, and $L_w^*(\mu, \mu_0)$ is the water-leaving radiance from VSLEAVE computed with unit transmittance. Here, $\mu_0$ is the solar zenith cosine, and $\mu$ any outgoing stream direction; we assume

azimuth-independence.

To find the coupling adjustment for $T^\downarrow(\mu_0)$, we an initial estimate $T_0^\downarrow(\mu_0)$ which could be the quantity in Eqn. B1 above; another value which we have tried is $T_0^\downarrow(\mu_0) = \frac{3}{2}T_{Direct}(\mu_0)$. With this starting value, we then have an adjusted water-leaving radiance $L_0(\mu, \mu_0) = L_w^*(\mu, \mu_0)\,T_0^\downarrow(\mu_0)$ which is then input to a Fourier-zero (azimuth independent) VLIDORT radiative transfer (RT) com-

putation. From this RT computation we then derive an updated total downwelling transmittance $T_1^\downarrow(\mu_0)$, which in turn provides an updated water-leaving input $L_1(\mu, \mu_0) = L_w^*(\mu, \mu_0)\,T_1^\downarrow(\mu_0)$. We repeat the Fourier-zero VLIDORT radiative transfer calculation with this new input, yielding a new result $T_2^\downarrow(\mu_0)$ for the transmittance, and a new water-leaving value $L_{w,2}(\mu, \mu_0)$. This iteration is stopped when the relative difference in the value of $T^\downarrow(\mu_0)$ between two iterations is less

than some small convergence criterion. We have found that convergence is rapid: typically only 3 iterations are needed for convergence at the level of $10^{-6}$.



It is not necessary to carry out a full Fourier calculation for every step. The discrete-ordinate homogeneous solutions and particular integrals do not depend on the surface-leaving radiance, and they need to be established just once from the initial Fourier-zero computation. Also, the complete discrete-ordinate solution is determined through the linear-algebra boundary value problem (BVP)

$\mathbf{Ax} = \mathbf{B}$, where matrix $\mathbf{A}$ is constructed entirely from the homogeneous solutions to the radiative transfer equation (RTE), $\mathbf{x}$ is the vector of unknown homogeneous-solution integration constants, and vector $\mathbf{B}$ is constructed from the layer particular integrals and also contains the surface boundary condition appropriate for water-leaving. Once the matrix inverse $\mathbf{A}^{-1}$ is found, the BVP solution is obtained through straightforward back-substitution: $\mathbf{x} = \mathbf{A}^{-1}\mathbf{B}$. Thus, the first guess for water

leaving input $L_0\left(\mu, \mu_0\right)$ will give rise to column vector $\mathbf{B}_0$, with corresponding solution $\mathbf{x}_0 = \mathbf{A}^{-1}\mathbf{B}_0$. From the discrete-ordinate solution based on $\mathbf{x}_0$, we then derive the next transmittance estimate $T_1^{\downarrow}\left(\mu_0\right)$, then form the next-guess water-leaving input $L_{w,1}\left(\mu, \mu_0\right)$ and associated column vector $\mathbf{B}_1$, from which we get the next solution $\mathbf{x}_1 = \mathbf{A}^{-1}\mathbf{B}_1$, and so on. All column vectors $\mathbf{B}_p$ are similar  only the surface-leaving entries are different. Thus the coupling adjustment is tantamount

to a series of back substitutions, and this represents very little extra computation load compared with the main radiative transfer equation, finding the inverse $\mathbf{A}^{-1}$. A three-iteration calculation is approximately 2% slower than a standard one.

Computation of the diffuse downwelling transmittance comes through the discrete-ordinate result:

$$T^{\downarrow}\left(\mu_0\right) = T_{diffuse}^{\downarrow}\left(\mu_0\right) + T_{direct}^{\downarrow}\left(\mu_0\right); \quad T_{diffuse}^{\downarrow} = \frac{2\pi}{\mu_0} \sum_{\alpha=1}^{n_d} I_{\alpha}^{\downarrow} \mu_{\alpha} c_{\alpha} \tag{B3}$$

$$I^{\downarrow} = \sum_{\alpha=1}^{n_d} L_{\alpha} Y_{\alpha}^{-} e^{-k_{\alpha}\Delta} + M_{\alpha} Y_{\alpha}^{+} + G^{\downarrow}(\mu_0). \tag{B4}$$

Here, $\mu_{\alpha}, c_{\alpha}, \alpha = 1, \ldots n_d$ are the discrete-ordinate quadrature values, $I^{\downarrow}$ is the downwelling intensity field at the surface expressed in terms of homogeneous solutions $Y_{\alpha}^{\pm}, k_{\alpha}$ in the lowest layer of the atmosphere, particular solutions $G^{\downarrow}\left(\mu_0\right)$ in that layer, and integration constants $L_{\alpha}, M_{\alpha}$ for that layer as determined from the BVP solution $\mathbf{x}_1 = \mathbf{A}^{-1}\mathbf{B}$. This flux computation does not require

any post-processing, nor any evaluations at other levels in the atmosphere.

*Data availability.* GLER will be available at https://aura.gesdisc.eosdis.nasa.gov/data/Aura_OMI_Level2/. The OMI Level 1 data used for calculations of GLER are available at https://aura.gesdisc.eosdis.nasa.gov/data/ Aura_OMI_Level1/ (last access: 11 April 2019). The OMI Level 2 Collection 3 data which include $NO_2$ and OMI pixel corner products are available at https://aura.gesdisc.eosdis.nasa.gov/data/

Aura_OMI_Level2/ (last access: 11 April 2019). The OMI $O_2$-$O_2$ cloud product can be provided upon request of the co-authors.





*Author contributions.* ZF led the paper and evaluation efforts. ZF, DH, AV, and RS wrote the paper. ZF and DH designed the GLER analysis. WQ carried out VLIDORT related simulations. NK and JJ provided guidance throughout the development of the manuscript. All authors contributed to the editing of the manuscript. RS and AS developed the VLIDORT code used for the BRDF and radiance computations.

*Competing interests.* The authors declare that they have no conflict of interest.

*Acknowledgements.* Funding for this work was provided by NASA through Aura core team funding as well as the Aura project and Aura Science Team and Atmospheric Composition Modeling and Analysis Program managed by Kenneth Jucks and Barry Lefer. We thank Patricia Castellanos and GMAO for providing us with the assimilated aerosol dataset. We thank David Antoine for useful discussions and providing data on the bidi-rectional aspects of water-leaving radiance. Additionally, we thank the MODIS and OMI teams for providing the calibrated data sets.





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
