# Peer review of "A geometry-dependent surface Lambertian-equivalent reflectivity product for UV/Vis retrievals: Part 2: Evaluation over open ocean"

_Atmospheric Measurement Techniques, 2019_

## Referee Comment (RC1) · Anonymous Referee #1 · 28 Jul 2019

General comments

The paper presents a lambertian-equivalent ocean surface reflectivity product for UV and visible wavelengths. The product is based on modeling of the outgoing radiation over Case 1 water scenes and takes into account satellite viewing geometry and ocean surface roughness. The product is a significant improvement over climatological lambertian reflectivity datasets used in many cloud, trace gas, and aerosol algorithms. A significant part of the paper is devoted to the evaluation of the product's performance in comparison with surface reflectance derived from the OMI instrument. The paper is well structured and the presentation is mostly clear. There are some issues with

[Figure]

Specific comments

Section 3.1: The analysis is based on data for a single month (January). It is evident form Fig. 9 that there a large seasonal varibility in the lambertian-equivalent ocean surface reflectivity which is due to the changing viewing geometry as well as changes in the input parameters. It is unclear if the numbers you quote in the section are applicable to other seasons or to the whole product. I suggest either adding data for June or redoing the analysis for a yearly (sub)sample. This is especially pertinent to any use of this product as a replacement for climatological datasets.

p. 10, l.1: Quote: "the cloud screening methods produce similar results with only small differences that do not impact the overall evaluation." To support the above statement I suggest adding a third table showing statistics for the left column of Figure 1.

top of the p. 11: I do not see "two main regions" in Fig.2 . It is not clear what two "distributions" the authors refer to as there is not clustering in the data. The range of 0.2-0.4 mentioned in the text appear to be arbitrary.

p. 15 Fig 8 and its analysis in the text: Figure 8 attempts to analyze the influence of aerosols based on the data from a single random orbit with a specific dependence of AOD on VZA. This analysis is obviously statistically insignificant and thus meaningless. A physical quantity like AOD should not depend on the observational geometry. Any such dependence is an indication of either a problem with the data or a lack of statistical power of the dataset. I suggest either removing this figure or redoing the analysis based on a better sample.

p.16, caption for Fig.9 and p.17 l. 8: Readers should not be expected to be familiar with the ONI row anomaly. Some discsussion and explanation of why a specific row was used is needed

p.17, l. 14: Quote: " in Fig. 10 there is a small downward trend in the difference be-tween GLER and OMI-derived LER of at most 0.005 LER. This may be related to the

downward drift in the OMI measurements" While it may be correct, the authors do not present enough evidence to support the statement and do not consider other possibilities. Trends in the auxiliary may be responsible. The authors used wind datasets from two different instruments with the switch occurring in the middle of the data series. How do the two datasets compare and could the switch affect the trend? In order to support their statement the author could adjust the calculations for the downward drift in the OMI measurements and see if they can reproduce the trend.

p.18 l.22 Wind speed and chlorophyll are two independent variables. Please describe how they were jointly perturbed to produce the results in Table 5.

Technical corrections

p.2, l. 5: due to p.2, l. 12: "angle" is missing after viewing p.8 Section 3.1 tables and graphs: The correspondence between tables and plots is not clear. Please state that Table 1 provides statistics for right column plots of Fig.1 in the caption. Same for Table 2 p.9, Fig. 1 caption's last sentence : Clarify that the left and right columns are for two cloud screening methods. p.20, l. 17: "combination of things" does not sound good; effects or factors?

---

## Referee Comment (RC2) · Anonymous Referee #2 · 19 Aug 2019

The current study derives the geometry-dependent surface Lambertian-equivalent reflectivity (GLER) using VLIODRT model by considering water leaving radiance for Case I water and polarized Cox-Munk water surface BRDF model. The calculation of GLER also makes use of satellite wind speed, MODIS ocean chlorophyll products, and GEOS-5 wind speed and direction datasets. Though validation against the OMI-derived LERs, the GLER data show consistent geometry dependency as well as low bias. Authors also investigated the sensitivity of GLER to potential error sources, including the presence of aerosols, wind speed, and chlorophyll concentrations. Overall, the study is reasonably designed and the paper is well written. It demonstrates that GLER is an improvements over climatological surface LER datasets, which is valuable

for implementation in the coming ocean missions such as the PACE. I only had a couple of comments listed as below.

Page 7, line 34-35 reads "We select sun glint scenes when the difference between the measured LER at 354 nm and 388 nm is less than -0.05." How is the threshold value chosen? Is there any reference or evidences indicating this number represents a good threshold? And why not define sun glint based on the co-scattering angle (or sun-glint angle)?

Page 17, line 3-5: "There is also a seasonal variation in GLER due to the changing viewing geometry of satellite measurements as the SZA changes through the year." Can the authors add more evidence to prove this statement? It seems to the sun glint may play an import role in the seasonal variation. I also curious if the seasonal variation is also related to any seasonal changes in wind speed or chlorophyl concentration. So it would be helpful if the time series for, sun glint angle, wind speed, and chlorophyll concentration are also provided (at least examined by the authors).

Ok. Continuing my last comment, the sensitivity analysis in section 3.5 indeed confirms that changes in chlorophyll concentration will not be able to cause the GLER seasonal variation.

Page 22, line 18: "lambda" is . –> "lambda" is the wavelength.

---

## Author Response (AR1)

Zachary Fasnacht
Science Systems and Applications, Inc.
10210 Greenbelt Rd. Lanham, MD 20706 USA

October 11, 2019

Dear Dr. Wang,

Please find enclosed an updated manuscript titled 'A geometry-dependent surface Lambertian-equivalent reflectivity product for UV/Vis retrievals: Part 2: Evaluation over open ocean' along with responses to reviewers comments. We made some changes to the manuscript in response to the reviewer's comments which are highlighted in a typeset latex difference document along with a list of changes provided here.

Both reviewers have asked some thought provoking questions in regards to the evaluation of GLER. We have addressed their concerns in making some minor changes and updates to the paper. We believe that doing so has only strengthened the conclusions made with this work. We greatly appreciate the reviewers' efforts and suggestions which have helped improve this paper.

Kind Regards,

Zachary Fasnacht

**List of Changes**

Title:

- Made a minor update changing *Part II.* to *Part 2:* in order to be consistent with the syntax of our first paper

Section 2.4 Ancillary Data for Water Model:

- Pg 5 Line 26: Fixed typo, AMSRE-E should be AMSR-E

Section 2.6 OMI Data and Selection Criteria

- Pg 7 Lines 19-24: We added a paragraph further explaining the OMI row anomaly as requested by reviewer #1
- Pg 7 Line 35 to Pg 8 Line 16: Reworded paragraph on sun glint classification to further clarify the methodology for determining sun glint affected pixels as there were some questions about this from reviewer #2

Section 3.1 Global Comparison of GLER and OMI-derived LER

- Per request from reviewer #1, we have added statistics for July 2006 to scatterplots and statistics tables to show that GLER captures the seasonality. Additionally we have added a screen on the solar zenith angle (ignore data above $70^{\circ}$) as high viewing angles can include cloud shadowing that decrease the OMI-derived LER.
- Added a new table to shows the statistics using the MODIS GCF per request from reviewer #1
- Corrected clerical error in Table #2 (now Table #3)
- Have made some changes to the text to address to acknowledge the additional information added to the scatterplots and statistics tables

Section 3.3 Simulating GLER with Aerosols

- Reviewer #1 noted that we only included a single orbit of data, so we have added all others orbits from April 10, 2006 into this analysis. The addition of more data had little impact on the results, but we have made note in the text that this is a specific test case as further analysis while likely be needed to formally quantify the aerosol impact on LER at these wavelengths.

Section 3.4 Inter-annual Variability of LER

- As requested by reviewer #2, we have added the wind speed, chlorophyll, and sun-glint angle to Figure 9 showing that the seasonal variability is simply caused by the seasonality in the sun glint

Section 3.5 Sensitivity to Chlorophyll and Wind Speed

- In response to a question from reviewer #1 about the method of the sensitivity test, we made minor changes to the wording of text to clarify how GLER was perturbed for the sensitivity analysis.

Appendix C Description of Look Up Tables

- We decided to include a brief paragraph in the appendix to describe the look up table approach that was used for processing GLER over the OMI mission

We greatly appreciate and thank the reviewers for their efforts related to this manuscript. They have provided important comments which have lead to several improvements in the paper. All comments from reviewers have been addressed below. *Reviewers' comments/questions below are denoted with italics,* responses are in plain text, **and additions to the manuscript are in bold**.

**Responses to Reviewer 1:**

*Section 3.1: The analysis is based on data for a single month (January). It is evident form Fig. 9 that there a large seasonal variability in the lambertian-equivalent ocean surface reflectivity which is due to the changing viewing geometry as well as changes in the input parameters. It is unclear if the numbers you quote in the section are applicable to other seasons or to the whole product. I suggest either adding data for June or redoing the analysis for a yearly (sub)sample. This is especially pertinent to any use of this product as a replacement for climatological datasets.*

*p. 10, l.1: Quote: "the cloud screening methods produce similar results with only small differences that do not impact the overall evaluation." To support the above statement I suggest adding a third table showing statistics for the left column of Figure 1.*

In response to the reviewer's first comments, we have included July 2006 results in the analysis for Section 3.1 and additionally added a table to show the statistics when MODIS GCF = 0.0. The reviewer makes a good comment about this quote which has been revised to note that the Raman based ECF leads to a better comparison than the MODIS GCF.

Pg 9 lines 3-5, we made a few changes to discuss the July 2006 data:

"As shown in Table 2, the GLER and the OMI-derived LER compare best **in January** at 388 nm where $R^2$ is 0.76 and the bias is 0.002 **with the Raman ECF (using the MODIS GCF $R^2$ is 0.60 and the bias is 0.007)."**

Pg 10 lines 1-6 were changed to account for the addition of the MODIS GCF in the comparison tables and the quote the reviewer noted above was changed

**"Overall the comparison is better using the Raman ECF cloud screen than when using the MODIS GCF. This is expected given that there is a 15 minute window between the** Aqua and Aura **overpass times in 2006 (becomes 7 minutes in 2009) leading to some change in cloud cover**. It is also worth noting since OMI has a wider swath than MODIS, cloud retrievals are not available **from MODIS** for pixels on the edge of the OMI swath (these pixels are not shown in Fig. 1). **For these reasons the Raman ECF will be used for cloud screening in the rest of the paper."**

*top of the p. 11: I do not see "two main regions" in Fig.2 . It is not clear what two "distributions" the authors refer to as there is not clustering in the data. The range of*
*0.2-0.4 mentioned in the text appear to be arbitrary.*

To make this paragraph more clear we have removed this statement as it has no impact on the results.

*p. 15 Fig 8 and its analysis in the text: Figure 8 attempts to analyze the influence of aerosols based on the data from a single random orbit with a specific dependence of AOD on VZA. This analysis is obviously statistically insignificant and thus meaningless. A physical quantity like AOD should not depend on the observational geometry. Any such dependence is an indication of either a problem with the data or a lack of statistical power of the dataset. I suggest either removing this figure or redoing the analysis based on a better sample.*

The reviewer was correct that we were showing only a limited sample so we have now included all orbits for April 10, 2006 over the Pacific Ocean. The original figure showed spatial dependence in AOD which caused a crosstrack dependence of AOD with higher AOD on the west side which is close to Asia. By adding the orbits for the rest of the day over the Pacific Ocean, this issue has been resolved (see Fig 8). This analysis is simply a small case study that shows the possible impact of aerosols. A more rigorous evaluation is needed to determine the exact quantitative impact of aerosols but is outside the scope of this paper and will be investigated further in the future.

We have reworded text in section 3.3 to make it more clear that this is simply a case study to show possible aerosol effects on the GLER results.

Pg 14 lines 6-8 the following word choice changes were made:

"Figure 5 shows the **MERRA-2 AOD and the LER change for orbit 9229 where AOD ranges** from around 0.05 in the South Pacific gyre to larger than 0.4 in the northern Pacific."

Pg 15 line 6-7, we updated the 466nm slope and RMSE for additional orbits in aerosol analysis

"The combination of these changes improves the slope from **1.16** before considering aerosols **at 466 nm** to **1.0** after aerosols are introduced."

Pg 15 lines 10-11, following sentence added to note the LER difference across the swath:
**"Figure 8 shows that aerosols increase GLER generally by 0.01-0.02, with the largest increase at large forward scattering angles."**

Pg 15, line 12, stylistic change was made to remove "*on the whole*"

Pg 15, line 16-17, stylistic change was made to change "*little to no change*" to "**small decrease in LER**"

Pg 16, lines 3-8, some text was reworded to emphasize that this is simply a case study

**"In this case study**, we note that an AOD of 0.1-0.15 **increased the LER by** as much as **0.01-0.02 at 466nm, with the largest increase being** in the forward scattering direction. At 354nm, however, similar AOD values **slightly decrease LER in the backward scatter** direction, but can increase LER by as much as 0.01 in the forward scattering direction. **While this analysis was for only a specific case study, we note that the aerosol contribution likely accounts for some of the difference between GLER and the OMI-derived LER."**

*p.16, caption for Fig.9 and p.17 l. 8: Readers should not be expected to be familiar with the OMI row anomaly. Some discussion and explanation of why a specific row was used is needed*

We have added the following text to section 2.6, pg 7, lines 14-20:

**"Beginning in mid-2007, OMI experienced an anomaly known as the "row anomaly" that has affected the L1b radiance data. There have been several impacts from the row anomaly including decreased radiances due to possible blockage, increased signal due sunlight being reflected into the instrument, a wavelength shift due to a change in the slit function, and earthshine radiances from outside the FOV that are reflected into the nadir port. The row anomaly is further explained in Schenkeveld et al. (2017). For this reason, after 2007 we focus only on rows 1-21, which are not affected by the row anomaly."**

*p.17, l. 14: Quote: " in Fig. 10 there is a small downward trend in the difference between GLER and OMI-derived LER of at most 0.005 LER. This may be related to the downward drift in the OMI measurements" While it may be correct, the authors do not present enough evidence to support the statement and do not consider other possibilities. Trends in the auxiliary may be responsible. The authors used wind datasets from two different instruments with the switch occurring in the middle of the data series. How do the two datasets compare and could the switch affect the trend? In order to support their statement the author could adjust the calculations for the downward drift in the OMI measurements and see if they can reproduce the trend.*

The reviewer raises an important point about other possible causes for the trend in the LER difference. We do note, however, that if the cause of the drift was the wind speed, the largest change would be at the longer wavelengths where the direct reflectance is more important and wind speed uncertainty is the greatest as shown in section 3.5.

Additionally, reviewer #2 has requested the inclusion of sun-glint angle, wind speed, and chlorophyll in Fig. 9. The change as noted in 2011 in the wind speeds is quite small and there is no apparent drift after this point. From 2016-2018, both

wind speed and chlorophyll appear quite consistent, but the GLER-OMI difference still appears to drift through these 3 years.

Attached we have included Fig 1. of the response in which we took Fig. 9 and de-seasonalized the data in order to better see the trends. As seen in Fig. 10, the trend in the LER difference at 354nm is only 0.005 and negligible at 466nm. Based on Fig. 1 of the response, at 354nm it appears that while GLER seems consistent, OMI LER does decrease starting around 2011/2012. We do note, there are some year to year changes in LER possibly due to phenomenon such as El Nino/La Nina, but such changes are captured both by GLER and OMI-derived LER.

[Figure]

Figure 1: De-seasonalized trend of GLER and OMI LER corresponding to Fig. 9 of the original manuscript. All data were screened the same as Fig. 9.

The general consensus is that the OMI drift is 1-1.5% radiance through the first 10 years of the OMI mission as noted by Schenkeveld et al. 2017. We see a drift of about 0.005 in LER at 354nm, which corresponds to around a 1% drift through the

OMI mission. The OMI team is currently working to assess the drift and will be applying a correction to the radiance data in the next version of L1b processing.

Pg 16 lines 23-26 were changed to be more precise about the current accepted rate of the OMI drift:

"We note that in Fig. 10 there is a small downward trend in the difference between GLER and OMI-derived LER of at most 0.005 in LER. **At 354nm a change of 0.005 LER corresponds to approximately 1% in TOA radiance which is close to the 1-1.5% TOA radiance degradation noted by Schenkeveld et al. 2017.**"

*p.18 l.22 Wind speed and chlorophyll are two independent variables. Please describe how they were jointly perturbed to produce the results in Table 5.*

We re-calculated GLER by perturbing the wind speed and chlorophyll in either direction by their assumed uncertainty. In the case of the combined perturbations, we simply perturbed each input in 4 possible directions (CHL high & WSP high; CHL low & WSP low; CHL high & WSP low; CHL low & WHP high) and then simply averaged the magnitude of the difference between the original LER and the adjusted LER. Additionally, we determined the maximum LER difference from all possible scenarios.

Pg 19 Lines 14-19 were changed (updated text in bold) to address this concern:

**"To determine the combined effect, we additionally calculated GLER perturbing both the wind speed and chlorophyll for the four possible combinations.** Table 6 shows the mean difference **from the combined sensitivity analysis** in GLER is similar to that obtained **by only perturbing the chlorophyll**. The maximum difference from the combined sensitivity test, however, is similar to that of the wind speed perturbation. This is because while the wind speed has a significant impact on sun glint, only a small fraction of OMI pixels are impacted by glint."

Technical corrections
*p.2, l. 5: due to*

*p.2, l. 12: "angle" is missing after viewing*

*p.8 Section 3.1 tables and graphs: The correspondence between tables and plots is not clear. Please state that*
*Table 1 provides statistics for right column plots of Fig.1 in the caption. Same for Table*
*2 p.9, Fig. 1 caption's last sentence : Clarify that the left and right columns are for two*
*cloud screening methods.*

*p.20, l. 17: "combination of things" does not sound good;*
*effects or factors?*

We greatly appreciate the reviewer noting these corrections and have made the corrections in the manuscript.

**Responses to Reviewer 2:**

*Page 7, line 34-35 reads "We select sun glint scenes when the difference between the measured LER at 354 nm and 388 nm is less than -0.05." How is the threshold value chosen? Is there any reference or evidences indicating this number represents a good threshold? And why not define sun glint based on the sun-glint angle (or sun-glint angle)?*

We changed the terminology of co-scattering angle to sun-glint angle because ultimately that is all that we meant by the co-scattering angle. Since sun-glint angle is more well known, we have taken the reviewers advice in making the change.

The reviewer raises some good questions about the methodology for choosing sun glint scenes. There appears to be some confusion with the text in the explanation of the sun glint scenes. We first focus on scenes where the sun-glint angle is less than 20 degrees and then additionally apply the LER difference screen. The reason for the LER difference screen is that clouds in the sun glint region are difficult to detect and therefore remove. Cloud fractions are typically biased high for sun glint regions making separation between clouds and sun glint challenging. By using the LER difference screen, we are able to more easily distinguish between clouds and sun glint.

Pg 7 Line 30-Pg 8 Line 4 was updated to address this concern:

"We compare cases with and without sun glint separately because the reflection of light in each case is quite different. For comparisons excluding sun glint scenes, we screen out data with a sun-glint angle of less than $20^{o}$ in which sun glint can occur. For the comparisons with sun glint, **while the sun-glint angle of $20^{o}$ is again used to choose the sun glint region, additional screening based on the spectral dependence of the measured LER is performed to remove clouds within the sun glint region. The reason for this is that cloud fraction retrievals are affected by sun glint. The** difference in LER occurs because of a spectrally dependent error in the underestimation of the Rayleigh scattering of diffuse light when one assumes a Lambertian ocean surface, when the reflectance is in fact specular. We select sun glint scenes when the difference between the measured LER at 354 nm and 388 nm is less than -0.05**. We note that some weaker sun glint has an LER difference that is not below this threshold, but here we focus on stronger glint that has no cloud contamination."**

We do note that due to a clerical error, a few values in Table 2 were incorrect, but this has since been corrected. The change is minor and does not have any impact on the final conclusions of the paper.

We did put some more thought to the threshold that was used for the LER difference in response to the reviewer's questions about it. We have done this by comparing the LER difference (354nm-388nm) from the OMI measurements with the absolute LER at 466nm from OMI. This analysis shown in Fig. 1 of the response shows two different distributions of data for cloud and sun glint. The data affected by clouds have a slope of nearly 0, whereas the sun glint data show a strong negative slope since the error in LER for sun glint is spectrally dependent. Using this shows that a threshold of -0.05 distinguishes sun glint from clouds. Below is the figure showing this comparison for January 2006 for data with a sun-glint angle less than 20°. The cutoff of -0.05 is shown in the solid black line with everything below it being included in the sun glint analysis.

[Figure]

Figure 1: Comparison of 466nm OMI-derived LER and the difference between 354nm and 466nm OMI-derived LER for January 2006.

Similarly in Fig. 2 of the response, we plotted the LER difference as a function of crosstrack position for January 2006 for sun-glint angles less than 20°. For OMI, sun glint typically occurs in rows 10-30. Here it also appears clear that a cutoff of -0.05 is effective to define the strong sun glint region.

[Figure]

Fig 2: LER difference plotted as a function of crosstrack for sun glint possible pixels defined as sun-glint angle less than 20 degrees

Finally, Fig. 3 of the response compares MODIS visible satellite imagery with OMI LER at 466nm and the LER difference from OMI to show that a threshold of -0.05 does a good job defining the stronger glint region.

[Figure]

Figure 3: OMI Delta R for an orbit on Jan 3, 2006 compared with the 466nm OMI-derived LER in the middle and MODIS visible satellite imagery on left

We note that this threshold is arbitrary in how it is chosen, but we feel that this provides the best evaluation of the sun glint data in avoiding contamination from clouds. We have refrained from including these figures in the paper at this time since the focus of this paper is not on the evaluation of our sun glint model.

*Page 17, line 3-5: "There is also a seasonal variation in GLER due to the changing viewing geometry of satellite measurements as the SZA changes through the year." Can the authors add more evidence to prove this statement? It seems to the sun glint may play an import role in the seasonal variation. I also curious if the seasonal variation is also related to any seasonal changes in wind speed or chlorophyl concentration. So it would be helpful if the time series for, sun glint angle, wind speed, and chlorophyll concentration are also provided (at least examined by the authors).*
*Ok. Continuing my last comment, the sensitivity analysis in section 3.5 indeed confirms that changes in chlorophyll concentration will not be able to cause the GLER seasonal variation.*

Per the reviewers' request, we have added these additional data to Fig. 9 and show that the seasonality is mainly due to the sun-glint angle seasonality. This request is very beneficial as it also addresses a question by reviewer 1 asking whether the chlorophyll or wind speed measurements could cause the drift in the GLER – OMI-derived LER difference. This figure shows that there is not a similar drift in either of the measurements meaning that the drift in the differences is likely at least partly instrumental.

*Page 22, line 18: "lambda" is . –> "lambda" is the wavelength.*

We have made the change noted by the reviewer.

[revised manuscript text omitted]
}\left(\lambda, \text{Chl}\right) = \text{Chl} \cdot \text{a}_1\left(\lambda\right) \cdot \text{Chl}^{-\text{b}_1(\lambda)}, \tag{A2}$$

where Chl is the chlorophyll concentration and $\lambda$ is wavelength. The value at 300 nm is used for all $\lambda < 300$ nm. The second source (over the range 400-720 nm) is based on linear interpolation of two sets of coefficients $a_2(\lambda), b_2(\lambda)$ at 10 nm intervals (Lee et al., 2005). The absorption formula in this regime is given by

$$\alpha_{Ph}\left(\lambda, \text{Chl}\right) = \left[a_2\left(\lambda\right) + b_2\left(\lambda\right) ln\left(a_{440}\right)\right] a_{440}, \tag{A3}$$

where $a_{440} = 0.06 \cdot \text{Chl}^{0.65}$ (Morel and Maritorena, 2001). The value at 720 nm is used for all $\lambda > 720$ nm. The CDOM absorption is given by Morel and Maritorena (2001):

$$\alpha_{CDOM}\left(\lambda, \text{Chl}\right) = 0.2 \cdot \left(\alpha_w(440\,\text{nm}) + 0.06 \cdot \text{Chl}^{0.65}\right) exp\left[-0.014\left(\lambda - 440\right)\right], \tag{A4}$$

where $\alpha_w(440\,\text{nm}) = 0.00635$ and $\lambda$ is in nm. The complete absorption is then

$$\alpha_{TOT}\left(\lambda, \text{Chl}\right) = \alpha_W\left(\lambda\right) + \alpha_{Ph}\left(\lambda, \text{Chl}\right) + \alpha_{CDOM}\left(\lambda, \text{Chl}\right). \tag{A5}$$

We use the following formula for the backscattering coefficient, assuming it is half of the scattering coefficient for pure water Rayleigh scattering, as described in Morel et al. (2007):

$$b_W\left(\lambda\right) = 0.0028\left(\frac{420}{\lambda}\right)^{4.3}, \tag{A6}$$

with $\lambda$ in nm. This For the particulate matter backscattering coefficient, we use the following from Morel and Maritorena (2001):

$$b_{Ph}\left(\lambda, \text{Chl}\right) = b_{Pb}\left(\text{Chl}\right)\beta_{bbp}\left(\text{Chl}, \lambda\right) \tag{A7}$$

$$b_{Pb}\left(\text{Chl}\right) = 0.416C^{0.766}; \quad \beta_{bbp}\left(\text{Chl}, \lambda\right) = 0.002 + 0.01\left[0.5 - 0.25log_{10}\text{Chl}\right]\left(\frac{\lambda}{550}\right)^V, \tag{A8}$$

where the exponent $V = 0$ for Chl $> 2$, and $V = 0.5\left[log_{10}\text{Chl} - 0.3\right]$ for Chl $\leq 2$. The complete backscattering is then

$$b_{TOT}\left(\lambda, \text{Chl}\right) = b_W\left(\lambda\right) + b_{ph}\left(\lambda, \text{Chl}\right). \tag{A9}$$

In the original formulation of water-leaving radiance in VLIDORT, the following formula was used to obtain the basic ocean-surface reflectance (Morel and Gentili, 1992):

$$R\left(\text{Chl}, \lambda, \mu_0\right) = f\left(\mu_0\right) R_{TOT}\left(\lambda, \text{Chl}\right) \equiv f\left(\mu_0\right)\frac{b_{TOT}\left(\lambda, \text{Chl}\right)}{a_{TOT}\left(\lambda, \text{Chl}\right)} \tag{A10}$$

$$f\left(\lambda, \text{Chl}, \theta_0\right) = d_0 - d_1\eta - d_2\eta^2 + \left(d_3\eta - d_4\right)\mu_0; \qquad \eta = \frac{b_W\left(\lambda\right)}{b_{TOT}\left(\lambda, \text{Chl}\right)}. \tag{A11}$$

Here, $f(\lambda, \text{Chl}, \theta_0)$ is given with 5 constants $\{\ d_0, d_1, d_2, d_3, d_4\ \} = \{\ 0.6279,\ 0.0227,\ 0.0513,$ $0.2465,\ 0.3119\ \}$, and $\mu_0 = \cos(\theta_0)$ is the cosine of the solar zenith angle. In order to assign the water-leaving radiance, the complete reflectance term is given by

5 $$R^{'}\left(\text{Chl}, \lambda, \mu_0\right) = \frac{R\left(\text{Chl}, \lambda, \mu_0\right)}{1 - \omega R\left(\text{Chl}, \lambda, \mu_0\right)}. \tag{A12}$$

Here, albedo $\omega = 0.485$, using the value in Austin (1974). The isotropic water-leaving radiance is then obtained after passage through the air-ocean interface:

$$S_{iso}\left(\text{Chl}, \lambda, \mu_0\right) \approx \frac{\mu_0}{\pi} T_{Surf}\left(\theta_0\right) \frac{R^{'}\left(\text{Chl}, \lambda, \mu_0\right)}{|n_w|^2}. \tag{A13}$$

Here, $n_w$ is the relative refractive index of water to air. For the flat surface case, the air-water
10 boundary transmittance $T_{Surf}\left(\theta_0\right)$ is often set to 1.0. In practice we use Fresnel optics to compute this quantity; values are typically 0.96 or more, depending on the value of $\theta_0$. In the rough surface case, $T_{Surf}\left(\theta_0\right)$ may be computed using glitter calculations based on Gaussian probability wave-facet distributions characterized by wind-speed and direction.

The above formulation does not account for the atmospheric transmitted flux $T_{atm}\left(\theta_0\right)$ at the
15 ocean surface, a quantity which is propagated through the interface. In the previous formulation, the ratio $\frac{T_{atm}(\theta_0)}{Q}$ was made implicit in the factor $\frac{\mu_0}{\pi}$ appearing in Eq. ( A13). Also, we replace the $f\left(\lambda, \text{Chl}, \theta_0\right)$ calculation with the direction-dependent ratio $\rho \equiv f/Q$ from Morel and Gentili (1996); Morel et al. (2002). The water-leaving radiance is then:

$$S\left(\text{Chl}, \lambda, \theta_0, \mu, \varphi\right) = \mu_0 T_{atm}\left(\theta_0\right) T_{Surf}\left(\theta_0\right) \frac{R^*\left(\text{Chl}, \lambda, \theta_0, \mu, \varphi\right)}{|n_w|^2} \tag{A14}$$

20 $$R^*\left(\text{Chl}, \lambda, \theta_0, \mu, \varphi\right) = \frac{\rho\left(\text{Chl}, \lambda, \theta_0, \mu, \varphi\right) R_{TOT}\left(\lambda, \text{Chl}\right)}{1 - \omega f\left(\text{Chl}, \lambda, \theta_0\right) R_{TOT}\left(\lambda, \text{Chl}\right)}, \tag{A15}$$

where $R_{TOT}\left(\lambda, \text{Chl}\right)$ is as defined in Eq. ( A10), and $\rho$ is the ratio $f/Q$. We use a tabulated form of the ratio $f/Q$ in our calculations.

In order to obtain an isotropic surface leaving radiance, we derive a quantity $\bar{\rho}\left(\text{Chl}, \lambda, \theta_0\right)$ from the $f/Q$ tables by averaging over all outgoing zenith and relative azimuth angles, $\theta$ and $\phi$, then
25 interpolating linearly with wavelength $\lambda$, followed by cubic spline interpolation and linear interpolation with the solar angle cosine $\mu_0$ and with the logarithm of the chlorophyll concentration. Spline interpolation is necessary because we want smooth and continuous derivatives with respect to Chl when considering linearization, as discussed below. The quantity $\bar{\rho}\left(\text{Chl}, \lambda, \theta_0\right)$ then defines the isotropic water-leaving contribution through:

30 $$S_{iso}\left(\text{Chl}, \lambda, \theta_0\right) = \mu_0 T_{atm}\left(\theta_0\right) T_{Surf}\left(\theta_0\right) \frac{\bar{R}^*\left(\text{Chl}, \lambda, \theta_0\right)}{|n_w|^2} \tag{A16}$$

$$\bar{R}^* \left( \text{Chl}, \lambda, \theta_0 \right) = \frac{\bar{\rho} \left( \text{Chl}, \lambda, \theta_0 \right) R_{TOT} \left( \lambda, \text{Chl} \right)}{1 - \omega f \left( \text{Chl}, \lambda, \theta_0 \right) R_{TOT} \left( \lambda, \text{Chl} \right)}. \tag{A17}$$

The azimuth dependence is very weak in the $f/Q$ tables, and we have omitted this dependence in the surface leaving formulation. However, we can derive non-isotropic surface-leaving $f/Q$ values by interpolating table entries with the cosine of the outgoing angle $\mu$. The resulting table extractions

5 are then $\tilde{\rho}_v \left( \text{Chl}, \lambda, \theta_0, \mu_\text{d} \right)$ and $\tilde{\rho}_d \left( \text{Chl}, \lambda, \theta_0, \mu_\text{d} \right)$ for each viewing angle $\mu_v$ and discrete ordinate stream $\mu_d$; these quantities are azimuth-averaged. We then have

$$S_v \left( \text{Chl}, \lambda, \theta_0, \mu_\text{v} \right) = \mu_0 T_{atm} \left( \theta_0 \right) T_{Surf} \left( \theta_0 \right) \frac{R_v^* \left( \text{Chl}, \lambda, \theta_0, \mu_\text{v} \right)}{|n_w|^2} \tag{A18}$$

$$R_v^* \left( \text{Chl}, \lambda, \theta_0, \mu_\text{v} \right) = \frac{\tilde{\rho}_v \left( \text{Chl}, \lambda, \theta_0, \mu_\text{v} \
[revised manuscript text omitted]